# CO-Optimal Transport

**Ievgen Redko**[*]
Univ Lyon, UJM-Saint-Etienne, CNRS, UMR 5516
F-42023, Saint-Etienne
ievgen.redko@univ-st-etienne.fr

**Titouan Vayer**[*]
Univ. Bretagne-Sud, CNRS, IRISA
F-56000 Vannes
titouan.vayer@irisa.fr

**Rémi Flamary**[*]
École Polytechnique, CMAP, UMR 7641
F-91120 Palaiseau
remi.flamary@polytechnique.edu

**Nicolas Courty**[*]
Univ. Bretagne-Sud, CNRS, IRISA
F-56000 Vannes
nicolas.courty@irisa.fr

## Abstract

Optimal transport (OT) is a powerful geometric and probabilistic tool for finding correspondences and measuring similarity between two distributions. Yet, its original formulation relies on the existence of a cost function between the samples of the two distributions, which makes it impractical when they are supported on different spaces. To circumvent this limitation, we propose a novel OT problem, named COOT for CO-Optimal Transport, that simultaneously optimizes two transport maps between both samples and features, contrary to other approaches that either discard the individual features by focusing on pairwise distances between samples or need to model explicitly the relations between them. We provide a thorough theoretical analysis of our problem, establish its rich connections with other OT-based distances and demonstrate its versatility with two machine learning applications in heterogeneous domain adaptation and co-clustering/data summarization, where COOT leads to performance improvements over the state-of-the-art methods.

## 1  Introduction

The problem of comparing two sets of samples arises in many fields in machine learning, such as manifold alignment [1], image registration [2], unsupervised word and sentence translation [3] among others. When correspondences between the sets are known *a priori*, one can align them with a global transformation of the features, *e.g*, with the widely used *Procrustes analysis* [4, 5]. For unknown correspondences, other popular alternatives to this method include correspondence free manifold alignment procedure [6], soft assignment coupled with a Procrustes matching [7] or Iterative closest point and its variants for 3D shapes [8, 9].

When one models the considered sets of samples as empirical probability distributions, Optimal Transport (OT) framework provides a solution to find, without supervision, a soft-correspondence map between them given by an *optimal coupling*. OT-based approaches have been used with success in numerous applications such as embeddings' alignments [10, 11] and Domain Adaptation (DA) [12] to name a few. However, one important limit of using OT for such tasks is that the two sets are assumed to lie in the same space so that the cost between samples across them can be computed. This major drawback does not allow OT to handle correspondence estimation across heterogeneous spaces, preventing its application in problems such as, for instance, heterogeneous DA (HDA). To circumvent this restriction, one may rely on the Gromov-Wasserstein distance (GW) [13]: a non-convex quadratic OT problem that finds the correspondences between two sets of samples based on

---

[*]Authors contributed equally.

their pairwise intra-domain similarity (or distance) matrices. Such an approach was successfully applied to sets of samples that do not lie in the same Euclidean space, *e.g* for shapes [14], word embeddings [15] and HDA [16] mentioned previously. One important limit of GW is that it finds the samples' correspondences but discards the relations between the features by considering pairwise similarities only.

In this work, we propose a novel OT approach called CO-Optimal transport (COOT) that simultaneously infers the correspondences between the samples *and* the features of two arbitrary sets. Our new formulation includes GW as a special case, and has an extra-advantage of working with raw data directly without needing to compute, store and choose computationally demanding similarity measures required for the latter. Moreover, COOT provides a meaningful mapping between both instances and features across the two datasets thus having the virtue of being interpretable. We thoroughly analyze the proposed problem, derive an optimization procedure for it and highlight several insightful links to other approaches. On the practical side, we provide evidence of its versatility in machine learning by putting forward two applications in HDA and co-clustering where our approach achieves state-of-the-art results.

The rest of this paper is organized as follows. We introduce the COOT problem in Section 2 and give an optimization routine for solving it efficiently. In Section 3, we show how COOT is related to other OT-based distances and recover efficient solvers for some of them in particular cases. Finally, in Section 4, we present an experimental study providing highly competitive results in HDA and co-clustering compared to several baselines.

## 2 CO-Optimal transport (COOT)

**Notations.** The simplex histogram with $n$ bins is denoted by $\Delta_n = \{\mathbf{w} \in (\mathbb{R}_+)^n : \sum_{i=1}^n w_i = 1\}$. We further denote by $\otimes$ the tensor-matrix multiplication, *i.e.*, for a tensor $\mathbf{L} = (L_{i,j,k,l})$, $\mathbf{L} \otimes \mathbf{B}$ is the matrix $(\sum_{k,l} L_{i,j,k,l} B_{k,l})_{i,j}$. We use $\langle \cdot, \cdot \rangle$ for the matrix scalar product associated with the Frobenius norm $\| \cdot \|_F$ and $\otimes_K$ for the Kronecker product of matrices, *i.e.*, $\mathbf{A} \otimes_K \mathbf{B}$ gives a tensor $\mathbf{L}$ such that $L_{i,j,k,l} = A_{i,j} B_{k,l}$. We note $\mathbb{S}_n$ the group of permutations of $\{1, \cdots, n\} = [\![n]\!]$. Finally, we write $\mathbf{1}_d \in \mathbb{R}^d$ for a $d$-dimensional vector of ones and denote all matrices by upper-case bold letters (*i.e.*, $\mathbf{X}$) or lower-case Greek letters (*i.e.*, $\boldsymbol{\pi}$); all vectors are written in lower-case bold (*i.e.*, $\mathbf{x}$).

### 2.1 CO-Optimal transport optimization problem

We consider two datasets represented by matrices $\mathbf{X} = [\mathbf{x}_1, \ldots, \mathbf{x}_n]^T \in \mathbb{R}^{n \times d}$ and $\mathbf{X}' = [\mathbf{x}'_1, \ldots, \mathbf{x}'_{n'}]^T \in \mathbb{R}^{n' \times d'}$, where in general we assume that $n \neq n'$ and $d \neq d'$. In what follows, the rows of the datasets are denoted as *samples* and its columns as *features*. We endow the samples $(\mathbf{x}_i)_{i \in [\![n]\!]}$ and $(\mathbf{x}'_i)_{i \in [\![n']\!]}$ with weights $\mathbf{w} = [w_1, \ldots, w_n]^\top \in \Delta_n$ and $\mathbf{w}' = [w'_1, \ldots, w'_{n'}]^\top \in \Delta_{n'}$ that both lie in the simplex so as to define empirical distributions supported on $(\mathbf{x}_i)_{i \in [\![n]\!]}$ and $(\mathbf{x}'_i)_{i \in [\![n']\!]}$. In addition to these distributions, we similarly associate weights given by vectors $\mathbf{v} \in \Delta_d$ and $\mathbf{v}' \in \Delta_{d'}$ with features. Note that when no additional information is available about the data, all the weights' vectors can be set as uniform.

We define the CO-Optimal Transport problem as follows:

$$\min_{\substack{\boldsymbol{\pi}^s \in \Pi(\mathbf{w}, \mathbf{w}') \\ \boldsymbol{\pi}^v \in \Pi(\mathbf{v}, \mathbf{v}')}} \sum_{i,j,k,l} L(X_{i,k}, X'_{j,l}) \pi^s_{i,j} \pi^v_{k,l} = \min_{\substack{\boldsymbol{\pi}^s \in \Pi(\mathbf{w}, \mathbf{w}') \\ \boldsymbol{\pi}^v \in \Pi(\mathbf{v}, \mathbf{v}')}} \langle \mathbf{L}(\mathbf{X}, \mathbf{X}') \otimes \boldsymbol{\pi}^s, \boldsymbol{\pi}^v \rangle \quad (1)$$

where $L : \mathbb{R} \times \mathbb{R} \to \mathbb{R}_+$ is a divergence measure between 1D variables, $\mathbf{L}(\mathbf{X}, \mathbf{X}')$ is the $d \times d' \times n \times n'$ tensor of all pairwise divergences between the elements of $\mathbf{X}$ and $\mathbf{X}'$, and $\Pi(\cdot, \cdot)$ is the set of linear transport constraints defined for $\mathbf{w}, \mathbf{w}')$ as $\Pi(\mathbf{w}, \mathbf{w}') = \{\boldsymbol{\pi} | \boldsymbol{\pi} \geq \mathbf{0}, \boldsymbol{\pi} \mathbf{1}_{n'} = \mathbf{w}, \boldsymbol{\pi}^\top \mathbf{1}_n = \mathbf{w}'\}$ and similarly for $\mathbf{v}, \mathbf{v}'$. Note that problem (1) seeks for a simultaneous transport $\boldsymbol{\pi}^s$ between samples and a transport $\boldsymbol{\pi}^v$ between features across distributions. In the following, we write $\text{COOT}(\mathbf{X}, \mathbf{X}', \mathbf{w}, \mathbf{w}', \mathbf{v}, \mathbf{v}')$ (or $\text{COOT}(\mathbf{X}, \mathbf{X}')$ when it is clear from the context) to denote the objective value of the optimization problem (1).

Equation (1) can be also extended to the entropic regularized case favoured in the OT community for remedying the heavy computation burden of OT and reducing its sample complexity [17, 18, 19].

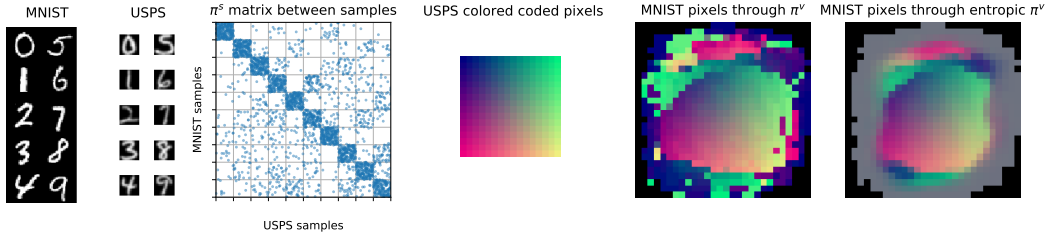

Figure 1: Illustration of COOT between MNIST and USPS datasets. **(left)** samples from MNIST and USPS data sets; **(center left)** Transport matrix $\boldsymbol{\pi}^s$ between samples sorted by class; **(center)** USPS image with pixels colored *w.r.t.* their 2D position; **(center right)** transported colors on MNIST image using $\boldsymbol{\pi}^v$, black pixels correspond to non-informative MNIST pixels always at 0; **(right)** transported colors on MNIST image using $\boldsymbol{\pi}^v$ with entropic regularization.

This leads to the following problem:

$$\min_{\boldsymbol{\pi}^s \in \Pi(\mathbf{w},\mathbf{w}'), \boldsymbol{\pi}^v \in \Pi(\mathbf{v},\mathbf{v}')} \langle \mathbf{L}(\mathbf{X},\mathbf{X}') \otimes \boldsymbol{\pi}^s, \boldsymbol{\pi}^v \rangle + \Omega(\boldsymbol{\pi}^s, \boldsymbol{\pi}^v) \tag{2}$$

where for $\epsilon_1, \epsilon_2 > 0$, the regularization term writes as $\Omega(\boldsymbol{\pi}^s, \boldsymbol{\pi}^v) = \epsilon_1 H(\boldsymbol{\pi}^s | \mathbf{w}\mathbf{w}'^T) + \epsilon_2 H(\boldsymbol{\pi}^v | \mathbf{v}\mathbf{v}'^T)$ with $H(\boldsymbol{\pi}^s | \mathbf{w}\mathbf{w}'^T) = \sum_{i,j} \log(\frac{\pi_{i,j}^s}{w_i w_j'}) \pi_{i,j}^s$ being the relative entropy. Note that similarly to OT [17] and GW [20], adding the regularization term can lead to a more robust estimation of the transport matrices but prevents them from being sparse.

**Illustration of COOT** In order to illustrate our proposed COOT method and to explain the intuition behind it, we solve the optimization problem (1) using the algorithm described in section 2.2 between two classical digit recognition datasets: MNIST and USPS. We choose these particular datasets for our illustration as they contain images of different resolutions (USPS is 16×16 and MNIST is 28×28) that belong to the same classes (digits between 0 and 9). Additionally, the digits are also slightly differently centered as illustrated on the examples in the left part of Figure 1. Altogether, this means that without specific pre-processing, the images do not lie in the same topological space and thus cannot be compared directly using conventional distances. We randomly select 300 images per class in each dataset, normalize magnitudes of pixels to $[0,1]$ and consider digit images as *samples* while each pixel acts as a *feature* leading to 256 and 784 features for USPS and MNIST respectively. We use uniform weights for $\mathbf{w}, \mathbf{w}'$ and normalize average values of each pixel for $\mathbf{v}, \mathbf{v}'$ in order to discard non-informative ones that are always equal to 0.

The result of solving problem (1) is reported in Figure 1. In the center-left part, we provide the coupling $\boldsymbol{\pi}^s$ between the samples, *i.e* the different images, sorted by class and observe that 67% of mappings occur between the samples from the same class as indicated by block diagonal structure of the coupling matrix. The coupling $\boldsymbol{\pi}^v$, in its turn, describes the relations between the features, *i.e* the pixels, in both domains. To visualize it, we color-code the pixels of the source USPS image and use $\boldsymbol{\pi}^v$ to transport the colors on a target MNIST image so that its pixels are defined as convex combinations of colors from the former with coefficients given by $\boldsymbol{\pi}^v$. The corresponding results are shown in the right part of Figure 1 for both the original COOT and its entropic regularized counterpart. From these two images, we can observe that colored pixels appear only in the central areas and exhibit a strong spatial coherency despite the fact that the geometric structure of the image is totally unknown to the optimization problem, as each pixel is treated as an independent variable. COOT has recovered a meaningful spatial transformation between the two datasets in a completely unsupervised way, different from trivial rescaling of images that one may expect when aligning USPS digits occupying the full image space and MNIST digits lying in the middle of it (for further evidence, other visualizations are given in the supplementary material).

**COOT as a billinear program** COOT is an indefinite Bilinear Program (BP) problem [21]: a special case of a Quadratic Program (QP) with linear constraints for which there exists an optimal solution lying on extremal points of the polytopes $\Pi(\mathbf{w},\mathbf{w}')$ and $\Pi(\mathbf{v},\mathbf{v}')$ [22, 23]. When $n = n', d = d'$ and weights $\mathbf{w} = \mathbf{w}' = \frac{\mathbf{1}_n}{n}, \mathbf{v} = \mathbf{v}' = \frac{\mathbf{1}_d}{d}$ are uniform, Birkhoff's theorem [24] states that the set of extremal points of $\Pi(\frac{\mathbf{1}_n}{n}, \frac{\mathbf{1}_n}{n})$ and $\Pi(\frac{\mathbf{1}_d}{d}, \frac{\mathbf{1}_d}{d})$ are the set of permutation matrices so that

---
**Algorithm 1** BCD for COOT
---
1: $\pi^s_{(0)} \leftarrow \mathbf{w}\mathbf{w}'^T, \pi^v_{(0)} \leftarrow \mathbf{v}\mathbf{v}'^T, k \leftarrow 0$
2: **while** $k <$ maxIt **and** $err > 0$ **do**
3:     $\boldsymbol{\pi}^v_{(k)} \leftarrow OT(\mathbf{v}, \mathbf{v}', \mathbf{L}(\mathbf{X}, \mathbf{X}') \otimes \boldsymbol{\pi}^s_{(k-1)})$ // OT problem on the samples
4:     $\boldsymbol{\pi}^s_{(k)} \leftarrow OT(\mathbf{w}, \mathbf{w}', \mathbf{L}(\mathbf{X}, \mathbf{X}') \otimes \boldsymbol{\pi}^v_{(k-1)})$ // OT problem on the features
5:     $err \leftarrow ||\boldsymbol{\pi}^v_{(k-1)} - \boldsymbol{\pi}^v_{(k)}||_F$
6:     $k \leftarrow k + 1$
---

there exists an optimal solution $(\boldsymbol{\pi}^s_*, \boldsymbol{\pi}^v_*)$ which transport maps are supported on two permutations $\sigma^s_*, \sigma^v_* \in \mathbb{S}_n \times \mathbb{S}_d$.

The BP problem is also related to the Bilinear Assignment Problem (BAP) where $\boldsymbol{\pi}^s$ and $\boldsymbol{\pi}^v$ are searched in the set of permutation matrices. The latter was shown to be NP-hard if $d = O(\sqrt[r]{n})$ for fixed $r$ and solvable in polynomial time if $d = O(\sqrt{\log(n)})$ [25]. In this case, we look for the best permutations of the rows and columns of our datasets that lead to the smallest cost. COOT provides a tight convex relaxation of the BAP by 1) relaxing the constraint set of permutations into the convex set of doubly stochastic matrices and 2) ensuring that two problems are equivalent, *i.e.*, one can always find a pair of permutations that minimizes (1), as explained in the paragraph above.

## 2.2 Properties of COOT

Finding a meaningful similarity measure between datasets is useful in many machine learning tasks as pointed out, *e.g* in [26]. To this end, COOT induces a distance between datasets $\mathbf{X}$ and $\mathbf{X}'$ and it vanishes *iff* they are the same up to a permutation of rows and columns as established below[2].

**Proposition 1** (COOT is a distance). *Suppose $L = |\cdot|^p, p \geq 1$, $n = n', d = d'$ and that the weights $\mathbf{w}, \mathbf{w}', \mathbf{v}, \mathbf{v}'$ are uniform. Then $COOT(\mathbf{X}, \mathbf{X}') = 0$ iff there exists a permutation of the samples $\sigma_1 \in \mathbb{S}_n$ and of the features $\sigma_2 \in \mathbb{S}_d$, s.t, $\forall i, k\ \mathbf{X}_{i,k} = \mathbf{X}'_{\sigma_1(i),\sigma_2(k)}$. Moreover, it is symmetric and satisfies the triangular inequality as long as $L$ satisfies the triangle inequality, i.e., $COOT(\mathbf{X}, \mathbf{X}'') \leq COOT(\mathbf{X}, \mathbf{X}') + COOT(\mathbf{X}', \mathbf{X}'')$.*

Note that in the general case when $n \neq n', d \neq d'$, positivity and triangle inequality still hold but $COOT(\mathbf{X}, \mathbf{X}') > 0$. Interestingly, our result generalizes the metric property proved in [27] for the election isomophism problem with this latter result being valid only for the BAP case (for a discussion on the connection between COOT and the work of [27], see supplementary material). Finally, we note that this metric property means that COOT can be used as a divergence in a large number of potential applications as, for instance, in generative learning [28].

## 2.3 Optimization algorithm and complexity

Even though solving COOT exactly may be NP-hard, in practice computing a solution can be done rather efficiently. To this end, we propose to use Block Coordinate Descent (BCD) that consists in iteratively solving the problem for $\boldsymbol{\pi}^s$ or $\boldsymbol{\pi}^v$ with the other kept fixed. Interestingly, this boils down to solving at each step a classical OT problem that requires $O(n^3 \log(n))$ operations with a network simplex algorithm. The pseudo-code of the proposed algorithm, known as the "mountain climbing procedure" [29], is given in Algorithm 1 and is guaranteed to decrease the loss after each update and so to converge within a finite number of iterations [23]. We also note that at each iteration one needs to compute the equivalent cost matrix $L(\mathbf{X}, \mathbf{X}') \otimes \boldsymbol{\pi}^{(\cdot)}$ which has a complexity of $O(ndn'd')$. However, one can reduce it using Proposition 1 from [20] for the case when $L$ is the squared Euclidean distance $|\cdot|^2$ or the Kullback-Leibler divergence. In this case, the overall computational complexity becomes $O(\min\{(n + n')dd' + n'^2n; (d + d')nn' + d'^2d\})$. In practice, we observed

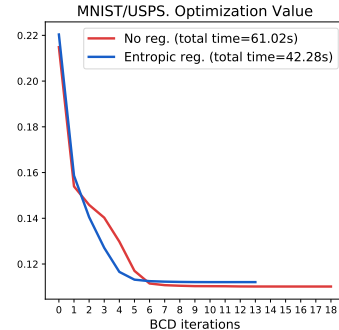

Figure 2: COOT loss during the BCD for the MNIST/USPS task.

in the numerical experiments that the BCD converges in few iterations (see *e.g.* Figure 2). We refer the interested reader to the supplementary material for further details. Finally, we can use the same BCD procedure for the entropic regularized version of COOT (2) where each iteration an entropic regularized OT problem can be solved efficiently using Sinkhorn's algorithm [17] with several possible improvements [18, 30, 31]. Note that this procedure can be easily adapted in the same way to include unbalanced OT problems [32] as well.

## 3   Relation with other OT distances

**Gromov-Wasserstein**   The COOT problem is defined for arbitrary matrices $\mathbf{X} \in \mathbb{R}^{n \times d}, \mathbf{X}' \in \mathbb{R}^{n' \times d'}$ and so can be readily used to compare pairwise similarity matrices between the samples $\mathbf{C} = (c(\mathbf{x}_i, \mathbf{x}_j)_{i,j}) \in \mathbb{R}^{n \times n}, \mathbf{C}' = (c'(\mathbf{x}'_k, \mathbf{x}'_l))_{k,l} \in \mathbb{R}^{n' \times n'}$ for some $c, c'$. To avoid redundancy, we use the term "similarity" for both similarity and distance functions in what follows. This situation arises in applications dealing with relational data, *e.g*, in a graph context [33] or deep metric alignement [34]. These problems have been successfully tackled recently using the Gromov-Wasserstein (GW) distance [13] which, given $\mathbf{C} \in \mathbb{R}^{n \times n}$ and $\mathbf{C}' \in \mathbb{R}^{n' \times n'}$, aims at solving:

$$GW(\mathbf{C}, \mathbf{C}', \mathbf{w}, \mathbf{w}') = \min_{\boldsymbol{\pi}^s \in \Pi(\mathbf{w}, \mathbf{w}')} \langle \mathbf{L}(\mathbf{C}, \mathbf{C}') \otimes \boldsymbol{\pi}^s, \boldsymbol{\pi}^s \rangle. \tag{3}$$

Below, we explicit the link between GW and COOT using a reduction of a concave QP to an associated BP problem established in [35] and show that they are equivalent when working with squared Euclidean distance matrices $\mathbf{C} \in \mathbb{R}^{n \times n}, \mathbf{C}' \in \mathbb{R}^{n' \times n'}$.

**Proposition 2.** *Let $L = |\cdot|^2$ and suppose that $\mathbf{C} \in \mathbb{R}^{n \times n}, \mathbf{C}' \in \mathbb{R}^{n' \times n'}$ are squared Euclidean distance matrices such that $\mathbf{C} = \mathbf{x}\mathbf{1}_n^T + \mathbf{1}_n\mathbf{x}^T - 2\mathbf{X}\mathbf{X}^T, \mathbf{C}' = \mathbf{x}'\mathbf{1}_{n'}^T + \mathbf{1}_{n'}\mathbf{x}'^T - 2\mathbf{X}'\mathbf{X}'^T$ with $\mathbf{x} = diag(\mathbf{X}\mathbf{X}^T), \mathbf{x}' = diag(\mathbf{X}'\mathbf{X}'^T)$. Then, the GW problem can be written as a concave quadratic program (QP) which Hessian reads $\mathbf{Q} = -4 * \mathbf{X}\mathbf{X}^T \otimes_K \mathbf{X}'\mathbf{X}'^T$.*

When working with arbitrary similarity matrices, COOT provides a lower-bound for GW and using Proposition 2 we can prove that both problems become equivalent in the Euclidean setting.

**Proposition 3.** *Let $\mathbf{C} \in \mathbb{R}^{n \times n}, \mathbf{C}' \in \mathbb{R}^{n' \times n'}$ be any symmetric matrices, then:*

$$COOT(\mathbf{C}, \mathbf{C}', \mathbf{w}, \mathbf{w}', \mathbf{w}, \mathbf{w}') \leq GW(\mathbf{C}, \mathbf{C}', \mathbf{w}, \mathbf{w}').$$

*The converse is also true under the hypothesis of Proposition 2. In this case, if $(\boldsymbol{\pi}^s_*, \boldsymbol{\pi}^v_*)$ is an optimal solution of (1), then both $\boldsymbol{\pi}^s_*, \boldsymbol{\pi}^v_*$ are solutions of (3). Conversely, if $\boldsymbol{\pi}^s_*$ is an optimal solution of (3), then $(\boldsymbol{\pi}^s_*, \boldsymbol{\pi}^s_*)$ is an optimal solution for (1) .*

Under the hypothesis of Proposition 2 we know that there exists an optimal solution for the COOT problem of the form $(\boldsymbol{\pi}_*, \boldsymbol{\pi}_*)$, where $\boldsymbol{\pi}_*$ is an optimal solution of the GW problem. This gives a conceptually very simple fixed-point procedure to compute an optimal solution of GW where one optimises over one coupling only and sets $\boldsymbol{\pi}^s_{(k)} = \boldsymbol{\pi}^v_{(k)}$ at each iteration of Algorithm 1. Interestingly enough, in the concave setting, these iterations are exactly equivalent to the Frank Wolfe algorithm described in [33] for solving GW. It also corresponds to a Difference of Convex Algorithm (DCA) [37, 38] where the concave function is approximated at each iteration by its linear majorization. When used for entropic regularized COOT, the resulting algorithm also recovers exactly the projected gradients iterations proposed in [20] for solving the entropic regularized version of GW. We refer the reader to the supplementary material for more details.

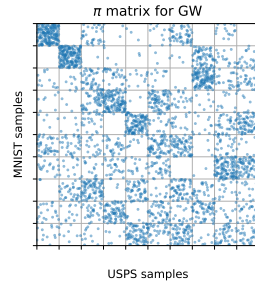

Figure 3: GW samples' coupling for MNIST-USPS task

To conclude, we would like to stress out that COOT is much more than a generalization of GW and that is for multiple reasons. First, it can be used on raw data without requiring to choose or compute the similarity matrices, that can be prohibitively costly, for instance, when dealing with shortest path distances in graphs, and to store them ($O(n^2 + n'^2)$ overhead). Second, it can take into account additional information given by feature weights $\mathbf{v}, \mathbf{v}'$ and provides an interpretable mapping between them across two heterogeneous datasets. Finally, contrary to GW, COOT is not invariant neither to feature rotations nor to the change of signs leading to a more informative samples' coupling when

| Domains | No-adaptation baseline | CCA | KCCA | EGW | SGW | COOT |
|---|---|---|---|---|---|---|
| C→W | 69.12±4.82 | 11.47±3.78 | 66.76±4.40 | 11.35±1.93 | 78.88±3.90 | **83.47**±2.60 |
| W→C | 83.00±3.95 | 19.59±7.71 | 76.76±4.70 | 11.00±1.05 | 92.41±2.18 | **93.65**±1.80 |
| W→W | 82.18±3.63 | 14.76±3.15 | 78.94±3.94 | 10.18±1.64 | 93.12±3.14 | **93.94**±1.84 |
| W→A | 84.29±3.35 | 17.00±12.41 | 78.94±6.13 | 7.24±2.78 | 93.41±2.18 | **94.71**±1.49 |
| A→C | 83.71±1.82 | 15.29±3.88 | 76.35±4.07 | 9.82±1.37 | 80.53±6.80 | **89.53**±2.34 |
| A→W | 81.88±3.69 | 12.59±2.92 | 81.41±3.93 | 12.65±1.21 | 87.18±5.23 | **92.06**±1.73 |
| A→A | 84.18±3.45 | 13.88±2.88 | 80.65±3.03 | 14.29±4.23 | 82.76±6.63 | **92.12**±1.79 |
| C→C | 67.47±3.72 | 13.59±4.33 | 60.76±4.38 | 11.71±1.91 | 77.59±4.90 | **83.35**±2.31 |
| C→A | 66.18±4.47 | 13.71±6.15 | 63.35±4.32 | 11.82±2.58 | 75.94±5.58 | **82.41**±2.79 |
| **Mean** | 78.00±7.43 | 14.65±2.29 | 73.77±7.47 | 11.12±1.86 | 84.65±6.62 | **89.47**±4.74 |
| **p-value** | <.001 | <.001 | <.001 | <.001 | <.001 | - |

Table 1: **Semi-supervised HDA** for $n_t = 3$ from Decaf to GoogleNet task.

compared to GW in some applications. One such example is given in the previous MNIST-USPS transfer task (Figure 1) for which the coupling matrix obtained via GW (given in Figure 3) exhibits important flaws in respecting class memberships when aligning samples.

**Invariant OT and Hierarchical OT** In [10], the authors proposed InvOT algorithm that aligns samples and learns a transformation between the features of two data matrices given by a linear map with a bounded Schatten p-norm. The authors further showed in [10, Lemma 4.3] that, under some mild assumptions, InvOT and GW lead to the same samples' couplings when cosine similarity matrices are used. It can be proved that, in this case, COOT is also equivalent to them both (see supplementary). However, note that InvOT is applicable under the strong assumption that $d = d'$ and provides only linear relations between the features, whereas COOT works when $d \neq d'$ and its feature mappings is sparse and more interpretable. InvOT was further used as a building block for aligning clustered datasets in [39] where the authors applied it as a divergence measure between the clusters, thus leading to an approach different from ours. Finally, in [40] the authors proposed a hierarchical OT distance as an OT problem with costs defined based on precomputed Wasserstein distances but with no global features' mapping, contrary to COOT that optimises two couplings of the features and the samples simultaneously.

## 4 Numerical experiments

In this section, we highlight two possible applications of COOT in a machine learning context: HDA and co-clustering. We consider these two particular tasks because 1) OT-based methods are considered as a strong baseline in DA; 2) COOT is a natural match for co-clustering as it allows for soft assignments of data samples and features to co-clusters.

### 4.1 Heterogeneous domain adaptation

In classification, domain adaptation problem arises when a model learned using a (source) domain $\mathbf{X}_s = \{\mathbf{x}_i^s\}_{i=1}^{N_s}$ with associated labels $\mathbf{Y}_s = \{\mathbf{y}_i^s\}_{i=1}^{N_s}$ is to be deployed on a related target domain $\mathbf{X}_t = \{\mathbf{x}_i^t\}_{i=1}^{N_t}$ where no or only few labelled data are available. Here, we are interested in the *heterogeneous* setting where the source and target data belong to different metric spaces. The most prominent works in HDA are based on Canonical Correlation Analysis [41] and its kernelized version and a more recent approach based on the Gromov-Wasserstein distance [16]. We investigate here the use of COOT for both *semi-supervised* HDA, where one has access to a small number $n_t$ of labelled samples per class in the target domain and *unsupervised* HDA with $n_t = 0$.

In order to solve the HDA problem, we compute $\text{COOT}(\mathbf{X}_s, \mathbf{X}_t)$ between the two domains and use the $\boldsymbol{\pi}^s$ matrix providing a transport/correspondence between samples (as illustrated in Figure 1) to estimate the labels in the target domain via label propagation [42]. Assuming uniform sample weights and one-hot encoded labels, a class prediction $\hat{\mathbf{Y}}_t$ in the target domain samples can be obtained by computing $\hat{\mathbf{Y}}_t = \boldsymbol{\pi}^s \mathbf{Y}_s$. When labelled target samples are available, we further prevent source samples to be mapped to target samples from a different class by adding a high cost in the cost matrix for every such source sample as suggested in [Sec. 4.2][12].

**Competing methods and experimental settings**    We evaluate COOT on *Amazon* (A), *Caltech-256* (C) and *Webcam* (W) domains from Caltech-Office dataset [43] with 10 overlapping classes between the domains and two different deep feature representations obtained for images from each domain using the Decaf [44] and GoogleNet [45] neural network architectures. In both cases, we extract the image representations as the activations of the last fully-connected layer, yielding respectively sparse 4096 and 1024 dimensional vectors. The heterogeneity comes from these two very different representations. We consider 4 baselines: CCA, its kernalized version KCCA [41] with a Gaussian kernel which width parameter is set to the inverse of the dimension of the input vector, EGW representing the entropic version of GW and SGW [16] that incorporates labelled target data into two regularization terms. For EGW and SGW, the entropic regularization term was set to 0.1, and the two other regularization hyperparameters for the semi-supervised case to $\lambda = 10^{-5}$ and $\gamma = 10^{-2}$ as done in [16**?** ]. We use COOT with entropic regularization on the feature mapping, with parameter $\epsilon_2 = 1$ in all experiments. For all OT methods, we use label propagation to obtain target labels as the maximum entry of $\hat{\mathbf{Y}}_t$ in each row. For all non-OT methods, classification was conducted with a k-nn classifier with $k = 3$. We run the experiment in a semi-supervised setting with $n_t = 3$, *i.e.*, 3 samples per class were labelled in the target domain. The baseline score is the result of classification by only considering labelled samples in the target domain as the training set. For each pair of domains, we selected 20 samples per class to form the learning sets. We run this random selection process 10 times and consider the mean accuracy of the different runs as a performance measure. In the presented results, we perform adaptation from Decaf to GoogleNet features, and report the results for $n_t \in \{0, 1, 3, 5\}$ in the opposite direction in the supplementary material.

**Results**    We first provide in Table 1 the results for the semi-supervised case. From it, we see that COOT surpasses all the other state-of-the-art methods in terms of mean accuracy. This result is confirmed by a $p$-value lower than 0.001 on a pairwise method comparison with COOT in a Wilcoxon signed rank test. SGW provides the second best result, while CCA and EGW have a less than average performance. Finally, KCCA performs better than the two latter methods, but still fails most of the time to surpass the no-adaptation baseline score given by a classifier learned on the available labelled target data. Results for the unsupervised case can be found in Table 2. This setting is rarely considered in the literature as unsupervised HDA is regarded as a very difficult problem. In this table, we do not provide scores for the no-adaptation baseline and SGW, as they require labelled data.

As one can expect, most of the methods fail in obtaining good classification accuracies in this setting, despite having access to discriminant feature representations. Yet, COOT succeeds in providing a meaningful mapping in some cases. The overall superior performance of COOT highlights its strengths and underlines the limits of other HDA methods. First, COOT does not depend on approximating empirical quantities from the data, contrary to CCA and KCCA that rely on the estimation of the cross-covariance matrix that is known to be flawed for

| Domains | CCA | KCCA | EGW | COOT |
|---|---|---|---|---|
| C→W | 14.20±8.60 | 21.30±15.64 | 10.55±1.97 | **25.50**±11.76 |
| W→C | 13.35±3.70 | 18.60±9.44 | 10.60±0.94 | **35.40**±14.61 |
| W→W | 10.95±2.36 | 13.25±6.34 | 10.25±2.26 | **37.10**±14.57 |
| W→A | 14.25±8.14 | 23.00±22.95 | 9.50±2.47 | **34.25**±13.03 |
| A→C | 11.40±3.23 | 11.50±9.23 | 11.35±1.38 | **17.40**±8.86 |
| A→W | 19.65±17.85 | 28.35±26.13 | 11.60±1.30 | **30.95**±18.19 |
| A→A | 11.75±1.82 | 14.20±4.78 | 13.10±2.35 | **42.85**±17.65 |
| C→C | 12.00±4.69 | 14.95±6.79 | 12.90±1.46 | **42.85**±18.44 |
| C→A | 15.35±6.30 | 23.35±17.61 | 12.95±2.63 | **33.25**±15.93 |
| **Mean** | 13.66±2.55 | 18.72±5.33 | 11.42±1.24 | **33.28**±7.61 |
| **p-value** | <.001 | <.001 | <.001 | - |

Table 2: **Unsupervised HDA** for $n_t = 0$ from Decaf to GoogleNet task.

high-dimensional data with few samples [46]. Second, COOT takes into account the features of the raw data that are more informative than the pairwise distances used in EGW. Finally, COOT avoids the sign invariance issue discussed previously that hinders GW's capability to recover classes without supervision as illustrated for the MNIST-USPS problem before.

## 4.2    Co-clustering and data summarization

While traditional clustering methods present an important discovery tool for data analysis, they discard the relationships that may exist between the features that describe the data samples. This idea is the cornerstone of *co-clustering* [47] where given a data matrix $\mathbf{X} \in \mathbb{R}^{n \times d}$ and the number of samples (rows) and features (columns) clusters denoted by $g \leq n$ and $m \leq d$, respectively, we seek to find $\mathbf{X}_c \in \mathbb{R}^{g \times m}$ that summarizes $\mathbf{X}$ in the best way possible.

**COOT-clustering**    We look for $\mathbf{X}_c$ which is as close as possible to the original $\mathbf{X}$ *w.r.t* COOT by solving $\min_{\mathbf{X}_c} \mathrm{COOT}(\mathbf{X}, \mathbf{X}_c) = \min_{\boldsymbol{\pi}^s, \boldsymbol{\pi}^v, \mathbf{X}_c} \langle \mathbf{L}(\mathbf{X}, \mathbf{X}_c) \otimes \boldsymbol{\pi}^s, \boldsymbol{\pi}^v \rangle$ with entropic regularization. More precisely, we set $\mathbf{w}, \mathbf{w}', \mathbf{v}, \mathbf{v}'$ as uniform, initialize $\mathbf{X}_c$ with random values and apply the BCD

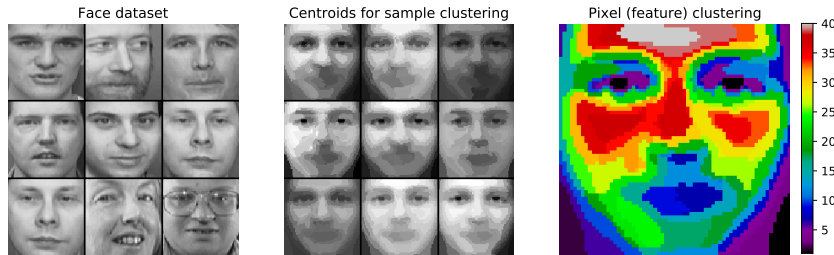

| Face dataset | Centroids for sample clustering | Pixel (feature) clustering |

Figure 4: Co-clustering with COOT on the Olivetti faces dataset. **(left)** Example images from the dataset, **(center)** centroids estimated by COOT **(right)** clustering of the pixels estimated by COOT where each color represents a cluster.

| Data set | Algorithms | | | | | | | | | |
|---|---|---|---|---|---|---|---|---|---|---|
| | K-means | NMF | DKM | Tri-NMF | GLBM | ITCC | RBC | CCOT | CCOT-GW | COOT |
| D1 | $.018 \pm .003$ | $.042 \pm .037$ | $.025 \pm .048$ | $.082 \pm .063$ | $.021 \pm .011$ | $.021 \pm .001$ | $.017 \pm .045$ | $.018 \pm .013$ | $.004 \pm .002$ | **0** |
| D2 | $.072 \pm .044$ | $.083 \pm .063$ | $.038 \pm .000$ | $.052 \pm .065$ | $.032 \pm .041$ | $.047 \pm .042$ | $.039 \pm .052$ | $.023 \pm .036$ | $.011 \pm .056$ | $\mathbf{.009 \pm 0.04}$ |
| D3 | – | – | $.310 \pm .000$ | – | $.262 \pm .022$ | $.241 \pm .031$ | – | $.031 \pm .027$ | $\mathbf{.008 \pm .001}$ | $.04 \pm .05$ |
| D4 | $.126 \pm .038$ | – | $.145 \pm .082$ | – | $.115 \pm .047$ | $.121 \pm .075$ | $.102 \pm .071$ | $.093 \pm .032$ | $.079 \pm .031$ | $\mathbf{0.068 \pm 0.04}$ |

Table 3: Mean ($\pm$ standard-deviation) of the co-clustering error (CCE) obtained for all configurations. "-" indicates that the algorithm cannot find a partition with the requested number of co-clusters. All the baselines results (first 9 columns) are from [48].

algorithm over $(\boldsymbol{\pi}^s, \boldsymbol{\pi}^v, \mathbf{X}_c)$ by alternating between the following steps: 1) obtain $\boldsymbol{\pi}^s$ and $\boldsymbol{\pi}^v$ by solving $\text{COOT}(\mathbf{X}, \mathbf{X}_c)$; 2) set $\mathbf{X}_c$ to $gm\boldsymbol{\pi}^{s\top}\mathbf{X}\boldsymbol{\pi}^v$. This second step of the procedure is a least-square estimation when $L = |\cdot|^2$ and corresponds to minimizing the COOT objective *w.r.t.* $\mathbf{X}_c$. In practice, we observed that few iterations of this procedure are enough to ensure the convergence. Once solved, we use the soft assignments provided by coupling matrices $\boldsymbol{\pi}^s \in \mathbb{R}^{n \times g}, \boldsymbol{\pi}^v \in \mathbb{R}^{d \times m}$ to partition data points and features to clusters by taking the index of the maximum element in each row of $\boldsymbol{\pi}^s$ and $\boldsymbol{\pi}^v$, respectively.

**Simulated data** We follow [48] where four scenarios with different number of co-clusters, degrees of separation and sizes were considered (for details, see the supplementary materials). We choose to evaluate COOT on simulated data as it provides us with the ground-truth for feature clusters that are often unavailable for real-world data sets. As in [48], we use the same co-clustering baselines including ITCC [49], Double K-Means (DKM) [50], Orthogonal Nonnegative Matrix Tri-Factorizations (ONTMF) [51], the Gaussian Latent Block Models (GLBM) [52] and Residual Bayesian Co-Clustering (RBC) [53] as well as the K-means and NMF run on both modes of the data matrix, as clustering baseline. The performance of all methods is measured using the co-clustering error (CCE) [54]. For all configurations, we generate 100 data sets and present the mean and standard deviation of the CCE over all sets for all baselines in Table 3. Based on these results, we see that our algorithm outperforms all the other baselines on D1, D2 and D4 data sets, while being behind CCOT-GW proposed by [48] on D3. This result is rather strong as our method relies on the original data matrix, while CCOT-GW relies on its kernel representation and thus benefits from the non-linear information captured by it. Finally, we note that while both competing methods rely on OT, they remain very different as CCOT-GW approach is based on detecting the positions and the number of jumps in the scaling vectors of GW entropic regularized solution, while our method relies on coupling matrices to obtain the partitions.

**Olivetti Face dataset** As a first application of COOT for the co-clustering problem on real data, we propose to run the algorithm on the well known Olivetti faces dataset [55].

We take 400 images normalized between 0 and 1 and run our algorithm with $g = 9$ image clusters and $m = 40$ feature (pixel) clusters. As before, we consider the empirical distributions supported on images and features, respectively. The resulting reconstructed image's clusters are given in Figure 4 and the pixel clusters are illustrated in its rightmost part. We can see that despite the high variability in the data set, we still manage to recover detailed centroids, whereas L2-based clustering such as standard NMF or k-means based on $\ell_2$ norm cost function are known to provide blurry estimates in this case. Finally, as in the MNIST-USPS example, COOT recovers spatially localized pixel clusters with no prior information about the pixel relations.

| M1 | M20 |
| --- | --- |
| Shawshank Redemption (1994) | Police Story 4: Project S (Chao ji ji hua) (1993) |
| Schindler's List (1993) | Eye of Vichy, The (Oeil de Vichy, L') (1993) |
| Casablanca (1942) | Promise, The (Versprechen, Das) (1994) |
| Rear Window (1954) | To Cross the Rubicon (1991) |
| Usual Suspects, The (1995) | Daens (1992) |

Table 4: Top 5 of movies in clusters M1 and M20. Average rating of the top 5 rated movies in M1 is 4.42, while for the M20 it is 1.

**MovieLens** We now evaluate our approach on the benchmark MOVIELENS-100K[3] data set that provides 100,000 user-movie ratings, on a scale of one to five, collected from 943 users on 1682 movies. The main goal of our algorithm here is to summarize the initial data matrix so that $\mathbf{X}_c$ reveals the blocks (co-clusters) of movies and users that share similar tastes. We set the number of user and film clusters to $g = 10$ and $m = 20$, respectively as in [56].

The obtained results provide the first movie cluster consisting of films with high ratings (3.92 on average), while the last movie cluster includes movies with very low ratings (1.92 on average). Among those, we show the 5 best/worst rated movies in those two clusters in Table 4. Overall, our algorithm manages to find a coherent co-clustering structure in MOVIELENS-100K and obtains results similar to those provided in [48, 56].

## 5    Discussion and conclusion

In this paper, we presented a novel optimal transport problem which aims at comparing distributions supported on samples belonging to different spaces. To this end, two optimal transport maps, one acting on the sample space, and the other on the feature space, are optimized to connect the two heterogeneous distributions. We provide several algorithms allowing to solve it in general and special cases and show its connections to other OT-based problems. We further demonstrate its usefulness and versatility on two difficult machine learning problems: heterogeneous domain adaptation and co-clustering/data summarization, where promising results were obtained. Numerous follow-ups of this work are expected. Beyond the potential applications of the method in various contexts, such as *e.g.* statistical matching, data analysis or even losses in deep learning settings, one immediate and intriguing question lies into the generalization of this framework in the continuous setting, and the potential connections to duality theory. This might lead to stochastic optimization schemes enabling large scale solvers for this problem.

### Acknowledgements

We thank Léo Gautheron, Guillaume Metzler and Raphaël Chevasson for proofreading the manuscript before the submission. This work benefited from the support from OATMIL ANR-17-CE23-0012 project of the French National Research Agency (ANR). This work has been supported by the French government, through the 3IA Côte d'Azur Investments in the Future project managed by the National Research Agency (ANR) with the reference number ANR-19-P3IA-0002. This action benefited from the support of the Chair "Challenging Technology for Responsible Energy" led by l'X – Ecole polytechnique and the Fondation de l'Ecole polytechnique, sponsored by TOTAL. We gratefully acknowledge the support of NVIDIA Corporation with the donation of the Titan X GPU used for this research.

## Broader impact

Despite its evident usefulness the problem of finding the correspondences between two datasets is rather general and may arise in many fields in machine learning. Consequently it is quite difficult to exhaustively state all the potential negative ethical impacts that may occur when using our method. As described in the paper, it could be used to solve the so-called election isomorphism problem [27] where one wants to find how similar are two elections based on the knowledge of votes and candidates. Although having these type of datasets seems unrealistic in modern democracies, using our approach

on this problem runs the risk of breaking some privacy standards by revealing precisely how the votes have been moved from one election to the other. Generally speaking, and when given access to two datasets with sensitive data, our method is able to infer correspondences between instances *and* features which could possibly lead to privacy issues for a malicious user. From a different perspective, the Optimal Transport framework is known to be quite computationally expensive and even recent improvements turns out to be super-linear in terms of the computational complexity. It is not an energy-free tool and in a time when carbon footprints must be drastically reduced, one should have in mind the potential negative impact that computationally demanding algorithms might have on the planet.

## Footnotes

[2]All proofs and theoretical results of this paper are detailed in the supplementary material.

[3]https://grouplens.org/datasets/movielens/100k/

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
