[Supplementary Material]

# Supplementary materials for paper:
# CO-Optimal Transport

**Ievgen Redko**
Univ Lyon, UJM-Saint-Etienne, CNRS, UMR 5516
F-42023, Saint-Etienne
ievgen.redko@univ-st-etienne.fr

**Titouan Vayer**
Univ. Bretagne-Sud, CNRS, IRISA
F-56000 Vannes
titouan.vayer@irisa.fr

**Rémi Flamary**
École Polytechnique, CMAP, UMR 7641
F-91120 Palaiseau
remi.flamary@polytechnique.edu

**Nicolas Courty**
Univ. Bretagne-Sud, CNRS, IRISA
F-56000 Vannes
nicolas.courty@irisa.fr

**Notations** We recall the notations of the paper. We consider two datasets represented by matrices $\mathbf{X} = [\mathbf{x}_1, \ldots, \mathbf{x}_n]^T \in \mathbb{R}^{n \times d}$ and $\mathbf{X}' = [\mathbf{x}'_1, \ldots, \mathbf{x}'_{n'}]^T \in \mathbb{R}^{n' \times d'}$. The rows of the datasets are denoted as *samples* and their columns as features. Let $\mu = \sum_{i=1}^{n} w_i \delta_{\mathbf{x}_i}$ and $\mu' = \sum_{i=1}^{n'} w'_i \delta_{\mathbf{x}'_i}$ be two empirical distributions related to the samples, where $\mathbf{x}_i \in \mathbb{R}^d$ and $\mathbf{x}'_i \in \mathbb{R}^{d'}$. We refer in the following to $\mathbf{w} = [w_1, \ldots, w_n]^\top$ and $\mathbf{w}' = [w'_1, \ldots, w'_{n'}]^\top$ as to sample weights vectors that both lie in the simplex ($\mathbf{w} \in \Delta_n$ and $\mathbf{w}' \in \Delta_{n'}$). In addition to them, we also introduce weights for the features that are stored on vectors $\mathbf{v} \in \Delta_d$ and $\mathbf{v}' \in \Delta_{d'}$. Finally, we let vec denote the column-stacking operator.

We define the CO-Optimal Transport (COOT) problem as follows:

$$\min_{\substack{\boldsymbol{\pi}^s \in \Pi(\mathbf{w},\mathbf{w}') \\ \boldsymbol{\pi}^v \in \Pi(\mathbf{v},\mathbf{v}')}} \sum_{i,j,k,l} L(X_{i,k}, X'_{j,l}) \boldsymbol{\pi}^s_{i,j} \boldsymbol{\pi}^v_{k,l} = \min_{\substack{\boldsymbol{\pi}^s \in \Pi(\mathbf{w},\mathbf{w}') \\ \boldsymbol{\pi}^v \in \Pi(\mathbf{v},\mathbf{v}')}} \langle \mathbf{L}(\mathbf{X}, \mathbf{X}') \otimes \boldsymbol{\pi}^s, \boldsymbol{\pi}^v \rangle, \qquad (1)$$

where $L : \mathbb{R} \times \mathbb{R} \to \mathbb{R}_+$ is a divergence measure between 1D variables, $\mathbf{L}$ is the tensor of all pairwise divergences between the elements of the matrices and $\Pi(\cdot, \cdot)$ is the set of linear transport constraints:

$$\Pi(\mathbf{w}, \mathbf{w}') = \{\boldsymbol{\pi} | \boldsymbol{\pi} \geq \mathbf{0}, \boldsymbol{\pi} \mathbf{1}_{n'} = \mathbf{w}, \boldsymbol{\pi}^\top \mathbf{1}_n = \mathbf{w}'\}.$$

The rest of the supplementary is organized as follows. After the MNIST-USPS illustration (Section 2 of the main paper), Section 1 presents the proof of Proposition 1 from the main paper and the computational complexity of calculating the value of the COOT problem as mentioned in Section 2.3 of the main paper. We provide the proofs for the equivalence of COOT to Gromov-Wassserstein distance (Propositions 2 and 3 from the main paper and algorithmic implications discussed after Proposition 3), InvOT and election isomorphism problem in Section 2. Finally, in Section 3, we provide additional experimental results for heterogeneous domain adaptation problem and precise the simulation details for the co-clustering task.

## Illustration on MNIST-USPS task

We provide a comparison between the coupling matrices obtained using GW and COOT on the MNIST-USPS problem from Section 2 of the main paper in Figure 1 and show the results of transporting the USPS samples to MNIST and vice versa using barycentric mapping in Figures 2 and 3.

Figure 1: Comparison between the coupling matrices obtained via GW and COOT on MNIST-USPS.

Figure 2: Linear mapping from USPS to MNIST using $\boldsymbol{\pi}^v$. (**First row**) Original USPS samples, (**Second row**) Samples resized to target resolution, (**Third row**) Samples mapped using $\boldsymbol{\pi}^v$, (**Fourth row**) Samples mapped using $\boldsymbol{\pi}^v$ with entropic regularization.

Figure 3: Linear mapping from MNIST to USPS using $\boldsymbol{\pi}^v$. (**First row**) Original MNIST samples, (**Second row**) Samples resized to target resolution, (**Third row**) Samples mapped using $\boldsymbol{\pi}^v$, (**Fourth row**) Samples mapped using $\boldsymbol{\pi}^v$ with entropic regularization.

# 1 Proofs from Section 2

## 1.1 Proof of Proposition 1

**Proposition 1** (COOT is a distance for $n = n', d = d'$)**.** *Suppose $L = |\cdot|^p, p \geq 1$, $n = n', d = d'$ and that the weights $\mathbf{w}, \mathbf{w}', \mathbf{v}, \mathbf{v}'$ are uniform. Then $COOT(\mathbf{X}, \mathbf{X}') = 0$ iff there exists a permutation of the samples $\sigma_1 \in \mathbb{S}_n$ and of the features $\sigma_2 \in \mathbb{S}_d$, s.t, $\forall i, k$ $\mathbf{X}_{i,k} = \mathbf{X}'_{\sigma_1(i),\sigma_2(k)}$. Moreover, in general for $n \neq n'$, $d \neq d'$ and potentially non uniform weights, it is symmetric and satisfies the triangle inequality as long as $L$ satisfies the triangle inequality $COOT(\mathbf{X}, \mathbf{X}'') \leq COOT(\mathbf{X}, \mathbf{X}') + COOT(\mathbf{X}', \mathbf{X}'')$.*

*Proof.* The symmetry follows from the definition of COOT. To prove the triangle inequality of COOT for arbitrary measures, we will use the gluing lemma (see [1]) which states the existence of couplings with a prescribed structure. Let $\mathbf{X} \in \mathbb{R}^{n \times d}, \mathbf{X}' \in \mathbb{R}^{n' \times d'}, \mathbf{X}'' \in \mathbb{R}^{n'' \times d''}$ associated with $\mathbf{w} \in \Delta_n, \mathbf{v} \in \Delta_d, \mathbf{w}' \in \Delta'_n, \mathbf{v}' \in \Delta'_d, \mathbf{w}'' \in \Delta''_n, \mathbf{v}'' \in \Delta''_d$. Without loss of generality, we can

suppose in the proof that all weights are different from zeros (otherwise we can consider $\tilde{w}_i = w_i$ if $w_i > 0$ and $\tilde{w}_i = 1$ if $w_i = 0$ see proof of Proposition 2.2 in [2])

Let $(\boldsymbol{\pi}_1^s, \boldsymbol{\pi}_1^v)$ and $(\boldsymbol{\pi}_2^s, \boldsymbol{\pi}_2^v)$ be two couples of optimal solutions for the COOT problems associated with $\text{COOT}(\mathbf{X}, \mathbf{X}', \mathbf{w}, \mathbf{w}', \mathbf{v}, \mathbf{v}')$ and $\text{COOT}(\mathbf{X}', \mathbf{X}'', \mathbf{w}', \mathbf{w}'', \mathbf{v}', \mathbf{v}'')$ respectively.

We define:

$$S_1 = \boldsymbol{\pi}_1^s \text{diag}\left(\frac{1}{\mathbf{w}'}\right) \boldsymbol{\pi}_2^s, \quad S_2 = \boldsymbol{\pi}_1^v \text{diag}\left(\frac{1}{\mathbf{v}'}\right) \boldsymbol{\pi}_2^v$$

Then, it is easy to check that $S_1 \in \Pi(\mathbf{w}, \mathbf{w}'')$ and $S_2 \in \Pi(\mathbf{v}, \mathbf{v}'')$ (see *e.g* Proposition 2.2 in [2]). We now show the following:

$$\text{COOT}(\mathbf{X}, \mathbf{X}'', \mathbf{w}, \mathbf{w}'', \mathbf{v}, \mathbf{v}'') \overset{*}{\leq} \langle \mathbf{L}(\mathbf{X}, \mathbf{X}'') \otimes S_1, S_2 \rangle = \langle \mathbf{L}(\mathbf{X}, \mathbf{X}'') \otimes [\boldsymbol{\pi}_1^s \text{diag}(\frac{1}{\mathbf{w}'})\boldsymbol{\pi}_2^s], [\boldsymbol{\pi}_1^v \text{diag}(\frac{1}{\mathbf{v}'})\boldsymbol{\pi}_2^v] \rangle$$

$$\overset{**}{\leq} \langle [\mathbf{L}(\mathbf{X}, \mathbf{X}') + \mathbf{L}(\mathbf{X}', \mathbf{X}'')] \otimes [\boldsymbol{\pi}_1^s \text{diag}(\frac{1}{\mathbf{w}'})\boldsymbol{\pi}_2^s], [\boldsymbol{\pi}_1^v \text{diag}(\frac{1}{\mathbf{v}'})\boldsymbol{\pi}_2^v] \rangle$$

$$= \langle \mathbf{L}(\mathbf{X}, \mathbf{X}') \otimes [\boldsymbol{\pi}_1^s \text{diag}(\frac{1}{\mathbf{w}'})\boldsymbol{\pi}_2^s], [\boldsymbol{\pi}_1^v \text{diag}(\frac{1}{\mathbf{v}'})\boldsymbol{\pi}_2^v] \rangle + \langle \mathbf{L}(\mathbf{X}', \mathbf{X}'') \otimes [\boldsymbol{\pi}_1^s \text{diag}(\frac{1}{\mathbf{w}'})\boldsymbol{\pi}_2^s], [\boldsymbol{\pi}_1^v \text{diag}(\frac{1}{\mathbf{v}'})\boldsymbol{\pi}_2^v] \rangle,$$

where in (*) we used the suboptimality of $S_1, S_2$ and in (**) the fact that $L$ satisfies the triangle inequality.

Now note that:

$$\langle \mathbf{L}(\mathbf{X}, \mathbf{X}') \otimes [\boldsymbol{\pi}_1^s \text{diag}(\frac{1}{\mathbf{w}'})\boldsymbol{\pi}_2^s], [\boldsymbol{\pi}_1^v \text{diag}(\frac{1}{\mathbf{v}'})\boldsymbol{\pi}_2^v] \rangle + \langle \mathbf{L}(\mathbf{X}', \mathbf{X}'') \otimes [\boldsymbol{\pi}_1^s \text{diag}(\frac{1}{\mathbf{w}'})\boldsymbol{\pi}_2^s], [\boldsymbol{\pi}_1^v \text{diag}(\frac{1}{\mathbf{v}'})\boldsymbol{\pi}_2^v] \rangle$$

$$= \sum_{i,j,k,l,e,o} L(X_{i,k}, X'_{e,o}) \frac{\boldsymbol{\pi}_{1\,i,e}^s \boldsymbol{\pi}_{2\,e,j}^s}{w'_e} \frac{\boldsymbol{\pi}_{1\,k,o}^v \boldsymbol{\pi}_{2\,o,l}^v}{v'_o} + \sum_{i,j,k,l,e,o} L(X'_{e,o}, X''_{j,l}) \frac{\boldsymbol{\pi}_{1\,i,e}^s \boldsymbol{\pi}_{2\,e,j}^s}{w'_e} \frac{\boldsymbol{\pi}_{1\,k,o}^v \boldsymbol{\pi}_{2\,o,l}^v}{v'_o}$$

$$\overset{*}{=} \sum_{i,k,e,o} L(X_{i,k}, X'_{e,o}) \boldsymbol{\pi}_{1\,i,e}^s \boldsymbol{\pi}_{1\,k,o}^v + \sum_{l,j,e,o} L(X'_{e,o}, X''_{j,l}) \boldsymbol{\pi}_{2\,e,j}^s \boldsymbol{\pi}_{2\,o,l}^v$$

where in (*) we used:

$$\sum_j \frac{\boldsymbol{\pi}_{2\,e,j}^s}{w'_e} = 1, \quad \sum_l \frac{\boldsymbol{\pi}_{2\,o,l}^v}{v'_o} = 1, \quad \sum_i \frac{\boldsymbol{\pi}_{1\,i,e}^s}{w'_e} = 1, \quad \sum_k \frac{\boldsymbol{\pi}_{1\,k,o}^v}{v'_o} = 1$$

Overall, from the definition of $\boldsymbol{\pi}_1^s, \boldsymbol{\pi}_1^v$ and $\boldsymbol{\pi}_2^s, \boldsymbol{\pi}_2^v$ we have:

$$\text{COOT}(\mathbf{X}, \mathbf{X}'', \mathbf{w}, \mathbf{w}'', \mathbf{v}, \mathbf{v}'') \leq \text{COOT}(\mathbf{X}, \mathbf{X}', \mathbf{w}, \mathbf{w}', \mathbf{v}, \mathbf{v}') + \text{COOT}(\mathbf{X}', \mathbf{X}'', \mathbf{w}', \mathbf{w}'', \mathbf{v}', \mathbf{v}'').$$

For the identity of indiscernibles, suppose that $n = n', d = d'$ and that the weights $\mathbf{w}, \mathbf{w}', \mathbf{v}, \mathbf{v}'$ are uniform. Suppose that there exists a permutation of the samples $\sigma_1 \in \mathbb{S}_n$ and of the features $\sigma_2 \in \mathbb{S}_d$, *s.t* $\forall i, k \in [\![n]\!] \times [\![d]\!]$, $\mathbf{X}_{i,k} = \mathbf{X}'_{\sigma_1(i),\sigma_2(k)}$. We define the couplings $\pi^s, \pi^v$ supported on the graphs of the permutations $\sigma_1, \sigma_2$ respectively, *i.e* $\pi^s = (Id \times \sigma_1)$ and $\pi^v = (Id \times \sigma_2)$. These couplings have the prescribed marginals and lead to a zero cost hence are optimal.

Conversely, as described in the paper, there always exists an optimal solution of (1) which lies on extremal points of the polytopes $\Pi(\mathbf{w}, \mathbf{w}')$ and $\Pi(\mathbf{v}, \mathbf{v}')$. When $n = n', d = d'$ and uniform weights are used, Birkhoffs theorem [3] states that the set of extremal points of $\Pi(\frac{\mathbf{1}_n}{n}, \frac{\mathbf{1}_n}{n})$ and $\Pi(\frac{\mathbf{1}_d}{d}, \frac{\mathbf{1}_d}{d})$ are the set of permutation matrices so there exists an optimal solution $(\pi_*^s, \pi_*^v)$ supported on $\sigma_*^s, \sigma_*^v$ respectively with $\sigma_*^s, \sigma_*^v \in \mathbb{S}_n \times \mathbb{S}_d$. Then, if $\text{COOT}(\mathbf{X}, \mathbf{X}') = 0$, it implies that $\sum_{i,k} L(X_{i,k}, X'_{\sigma_*^s(i),\sigma_*^v(k)}) = 0$. If $L = |\cdot|^p$ then $X_{i,k} = X'_{\sigma_*^s(i),\sigma_*^v(k)}$ which gives the desired result. If $n \neq n', d \neq d'$ the COOT cost is always strictly positive as there exists a strictly positive element outside the diagonal.

$\square$

## 1.2 Complexity of computing the value of COOT

As mentionned in [4], if $L$ can be written as $L(a, b) = f(a) + f(b) - h_1(a)h_2(b)$ then we have that

$$\mathbf{L}(\mathbf{X}, \mathbf{X}') \otimes \pi^s = \mathbf{C}_{\mathbf{X}, \mathbf{X}'} - h_1(\mathbf{X})\pi^s h_2(\mathbf{X}')^T,$$

where $\mathbf{C}_{\mathbf{X},\mathbf{X}'} = \mathbf{X}\mathbf{w}\mathbb{1}_{n'}^T + \mathbb{1}_n\mathbf{w}'^T\mathbf{X}'^T$ so that the latter can be computed in $O(ndd' + n'dd') = O((n+n')dd')$. To compute the final cost, we must also calculate the scalar product with $\boldsymbol{\pi}^v$ that can be done in $O(n'^2n)$ making the complexity of $\langle \mathbf{L}(\mathbf{X},\mathbf{X}') \otimes \boldsymbol{\pi}^s, \boldsymbol{\pi}^v \rangle$ equal to $O((n+n')dd' + n'^2n)$.

Finally, as the cost is symmetric *w.r.t* $\boldsymbol{\pi}^s, \boldsymbol{\pi}^v$, we obtain the overall complexity of $O(\min\{(n+n')dd' + n'^2n; (d+d')nn' + d'^2d\})$.

## 2 Proofs from Section 3

### 2.1 Equivalence between BAP and QAP

As pointed in [5], we can relate the solutions of a QAP and a BAP using the following theorem:

**Theorem 1.** *If* $\mathbf{Q}$ *is a positive semi-definite matrix, then problems:*

$$\begin{aligned} \max_{\mathbf{x}} f(\mathbf{x}) &= \mathbf{c}^T\mathbf{x} + \tfrac{1}{2}\mathbf{x}^T\mathbf{Q}\mathbf{x} \\ s.t. &\quad \mathbf{A}\mathbf{x} = \mathbf{b}, \ \mathbf{x} \geq 0 \end{aligned} \tag{2}$$

$$\begin{aligned} \max_{\mathbf{x},\mathbf{y}} g(\mathbf{x},\mathbf{y}) &= \tfrac{1}{2}\mathbf{c}^T\mathbf{x} + \tfrac{1}{2}\mathbf{c}^T\mathbf{y} + \tfrac{1}{2}\mathbf{x}^T\mathbf{Q}\mathbf{y} \\ s.t. &\quad \mathbf{A}\mathbf{x} = \mathbf{b}, \mathbf{A}\mathbf{y} = \mathbf{b}, \ \mathbf{x}, \mathbf{y} \geq 0 \end{aligned} \tag{3}$$

*are equivalent. More precisely, if* $\mathbf{x}^*$ *is an optimal solution for* (2)*, then* $(\mathbf{x}^*, \mathbf{x}^*)$ *is a solution for* (3) *and if* $(\mathbf{x}^*, \mathbf{y}^*)$ *is optimal for* (3)*, then both* $\mathbf{x}^*$ *and* $\mathbf{y}^*$ *are optimal for* (2)*.*

*Proof.* This proof follows the proof of Theorem 2.2 in [5]. Let $\mathbf{z}^*$ be optimal for (2) and $(\mathbf{x}^*, \mathbf{y}^*)$ be optimal for (3). Then, by definition, for all $\mathbf{x}$ satisfying the constraints of (2), $f(\mathbf{z}^*) \geq f(\mathbf{x})$. In particular, $f(\mathbf{z}^*) \geq f(\mathbf{x}^*) = g(\mathbf{x}^*, \mathbf{x}^*)$ and $f(\mathbf{z}^*) \geq f(\mathbf{y}^*) = g(\mathbf{y}^*, \mathbf{y}^*)$. Also, $g(\mathbf{x}^*, \mathbf{y}^*) \geq \max_{\mathbf{x},\mathbf{x} \text{ s.t } \mathbf{A}\mathbf{x}=\mathbf{b},\mathbf{x}\geq0} g(\mathbf{x}, \mathbf{x}) = f(\mathbf{z}^*)$.

To prove the theorem, it suffices to prove that

$$f(\mathbf{y}^*) = f(\mathbf{x}^*) = g(\mathbf{x}^*, \mathbf{y}^*) \tag{4}$$

since, in this case, $g(\mathbf{x}^*, \mathbf{y}^*) = f(\mathbf{x}^*) \geq f(\mathbf{z}^*)$ and $g(\mathbf{x}^*, \mathbf{y}^*) = f(\mathbf{y}^*) \geq f(\mathbf{z}^*)$.

Let us prove (4). Since $(\mathbf{x}^*, \mathbf{y}^*)$ is optimal, we have:

$$0 \leq g(\mathbf{x}^*, \mathbf{y}^*) - g(\mathbf{x}^*, \mathbf{x}^*) = \frac{1}{2}\mathbf{c}^T(\mathbf{y}^* - \mathbf{x}^*) + \frac{1}{2}\mathbf{x}^{*T}\mathbf{Q}(\mathbf{y}^* - \mathbf{x}^*)$$

$$0 \leq g(\mathbf{x}^*, \mathbf{y}^*) - g(\mathbf{y}^*, \mathbf{y}^*) = \frac{1}{2}\mathbf{c}^T(\mathbf{x}^* - \mathbf{y}^*) + \frac{1}{2}\mathbf{y}^{*T}\mathbf{Q}(\mathbf{x}^* - \mathbf{y}^*).$$

By adding these inequalities we obtain:

$$(\mathbf{x}^* - \mathbf{y}^*)^T\mathbf{Q}(\mathbf{x}^* - \mathbf{y}^*) \leq 0.$$

Since $\mathbf{Q}$ is positive semi-definite, this implies that $\mathbf{Q}(\mathbf{x}^* - \mathbf{y}^*) = 0$. So, using previous inequalities, we have $\mathbf{c}^T(\mathbf{x}^* - \mathbf{y}^*) = 0$, hence $g(\mathbf{x}^*, \mathbf{y}^*) = g(\mathbf{x}^*, \mathbf{x}^*) = g(\mathbf{y}^*, \mathbf{y}^*)$ as required.

Note also that this result holds when we add a constant term to the cost function. $\qquad\square$

### 2.2 Proofs of Propositions 2 and 3

We now prove all the theorems from Section 3 from the main paper. We first recall the GW problem for two matrices $\mathbf{C}, \mathbf{C}'$:

$$GW(\mathbf{C}, \mathbf{C}', \mathbf{w}, \mathbf{w}') = \min_{\boldsymbol{\pi}^s \in \Pi(\mathbf{w},\mathbf{w}')} \langle L(\mathbf{C}, \mathbf{C}') \otimes \boldsymbol{\pi}^s, \boldsymbol{\pi}^s \rangle. \tag{5}$$

We will now prove the Proposition 2 in the main paper stated as follows.

**Proposition 2.** *Let* $L = |\cdot|^2$ *and suppose that* $\mathbf{C} \in \mathbb{R}^{n \times n}, \mathbf{C}' \in \mathbb{R}^{n' \times n'}$ *are squared Euclidean distance matrices such that* $\mathbf{C} = \mathbf{x}\mathbb{1}_n^T + \mathbb{1}_n\mathbf{x}^T - 2\mathbf{X}\mathbf{X}^T, \mathbf{C}' = \mathbf{x}'\mathbb{1}_{n'}^T + \mathbb{1}_{n'}\mathbf{x}'^T - 2\mathbf{X}'\mathbf{X}'^T$ *with* $\mathbf{x} = diag(\mathbf{X}\mathbf{X}^T), \mathbf{x}' = diag(\mathbf{X}'\mathbf{X}'^T)$*. Then, the GW problem can be written as a concave quadratic program (QP) which Hessian reads* $\mathbf{Q} = -4 * \mathbf{X}\mathbf{X}^T \otimes_K \mathbf{X}'\mathbf{X}'^T$*.*

This result is a consequence of the following lemma.

**Lemma 1.** *With previous notations and hypotheses, the GW problem can be formulated as:*

$$GW(\mathbf{C}, \mathbf{C}', \mathbf{w}, \mathbf{w}') = \min_{\boldsymbol{\pi}^s \in \Pi(\mathbf{w}, \mathbf{w}')} -4vec(\mathbf{M})^T vec(\boldsymbol{\pi}^s) - 8vec(\boldsymbol{\pi}^s)^T \mathbf{Q} vec(\boldsymbol{\pi}^s) + Cte$$

*with*

$$\mathbf{M} = \mathbf{x}\mathbf{x}'^T - 2\mathbf{x}\mathbf{w}'^T\mathbf{X}'\mathbf{X}'^T - 2\mathbf{X}\mathbf{X}^T\mathbf{w}\mathbf{x}'^T \text{ and } \mathbf{Q} = \mathbf{X}\mathbf{X}^T \otimes_K \mathbf{X}'\mathbf{X}'^T,$$

$$Cte = \sum_i \|\mathbf{x}_i - \mathbf{x}_j\|_2^4 \mathbf{w}_i \mathbf{w}_j + \sum_i \|\mathbf{x}'_i - \mathbf{x}'_j\|_2^4 \mathbf{w}'_i \mathbf{w}'_j - 4\mathbf{w}^T \mathbf{x}\mathbf{w}'^T \mathbf{x}'$$

*Proof.* Using the results in [4] for $L = |\cdot|^2$, we have $\mathbf{L}(\mathbf{C}, \mathbf{C}') \otimes \boldsymbol{\pi}^s = c_{\mathbf{C}, \mathbf{C}'} - 2\mathbf{C}\boldsymbol{\pi}^s\mathbf{C}'$ with $c_{\mathbf{C}, \mathbf{C}'} = (\mathbf{C})^2\mathbf{w}\mathbf{1}_{n'}^T + \mathbf{1}_n \mathbf{w}'^T(\mathbf{C}')^2$, where $(\mathbf{C})^2 = (\mathbf{C}_{i,j}^2)$ is applied element-wise.

We now have that

$$\langle \mathbf{C}\boldsymbol{\pi}^s\mathbf{C}', \boldsymbol{\pi}^s \rangle = \text{tr}\big[\boldsymbol{\pi}^{sT}(\mathbf{x}\mathbf{1}_n^T + \mathbf{1}_n\mathbf{x}^T - 2\mathbf{X}\mathbf{X}^T)\boldsymbol{\pi}^s(\mathbf{x}'\mathbf{1}_{n'}^T + \mathbf{1}_{n'}\mathbf{x}'^T - 2\mathbf{X}'\mathbf{X}'^T)\big]$$

$$= \text{tr}\big[(\boldsymbol{\pi}^{sT}\mathbf{x}\mathbf{1}_n^T + \mathbf{w}'\mathbf{x}^T - 2\boldsymbol{\pi}^{sT}\mathbf{X}\mathbf{X}^T)(\boldsymbol{\pi}^s\mathbf{x}'\mathbf{1}_{n'}^T + \mathbf{w}\mathbf{x}'^T - 2\boldsymbol{\pi}^s\mathbf{X}'\mathbf{X}'^T)\big]$$

$$= \text{tr}\big[\boldsymbol{\pi}^{sT}\mathbf{x}\mathbf{w}'^T\mathbf{x}'\mathbf{1}_{n'}^T + \boldsymbol{\pi}^{sT}\mathbf{x}\mathbf{x}'^T - 2\boldsymbol{\pi}^{sT}\mathbf{x}\mathbf{w}'^T\mathbf{X}'\mathbf{X}'^T + \mathbf{w}'\mathbf{x}^T\boldsymbol{\pi}^s\mathbf{x}'\mathbf{1}_{n'}^T + \mathbf{w}'\mathbf{x}^T\mathbf{w}\mathbf{x}'^T - 2\mathbf{w}'\mathbf{x}^T\boldsymbol{\pi}^s\mathbf{X}'\mathbf{X}'^T$$

$$- 2\boldsymbol{\pi}^{sT}\mathbf{X}\mathbf{X}^T\boldsymbol{\pi}^s\mathbf{x}'\mathbf{1}_{n'}^T - 2\boldsymbol{\pi}^{sT}\mathbf{X}\mathbf{X}^T\mathbf{w}\mathbf{x}'^T + 4\boldsymbol{\pi}^{sT}\mathbf{X}\mathbf{X}^T\boldsymbol{\pi}^s\mathbf{X}'\mathbf{X}'^T\big]$$

$$\overset{*}{=} \text{tr}\big[\boldsymbol{\pi}^{sT}\mathbf{x}\mathbf{w}'^T(\mathbf{x}'\mathbf{1}_{n'}^T + \mathbf{1}_{n'}\mathbf{x}'^T) + \boldsymbol{\pi}^{sT}\mathbf{x}\mathbf{x}'^T + \mathbf{w}'\mathbf{x}^T\mathbf{w}\mathbf{x}'^T - 2\boldsymbol{\pi}^{sT}\mathbf{x}\mathbf{w}'^T\mathbf{X}'\mathbf{X}'^T - 2\mathbf{w}'\mathbf{x}^T\boldsymbol{\pi}^s\mathbf{X}'\mathbf{X}'^T$$

$$- 2\boldsymbol{\pi}^{sT}\mathbf{X}\mathbf{X}^T\boldsymbol{\pi}^s\mathbf{x}'\mathbf{1}_{n'}^T - 2\boldsymbol{\pi}^{sT}\mathbf{X}\mathbf{X}^T\mathbf{w}\mathbf{x}'^T + 4\boldsymbol{\pi}^{sT}\mathbf{X}\mathbf{X}^T\boldsymbol{\pi}^s\mathbf{X}'\mathbf{X}'^T\big],$$

where in (*) we used:

$$\text{tr}(\mathbf{w}'\mathbf{x}^T\boldsymbol{\pi}^s\mathbf{x}'\mathbf{1}_{n'}^T) = \text{tr}(\mathbf{x}'\mathbf{1}_{n'}^T\mathbf{w}'\mathbf{x}^T\boldsymbol{\pi}^s) = \text{tr}(\boldsymbol{\pi}^{sT}\mathbf{x}\mathbf{w}'^T\mathbf{1}_{n'}\mathbf{x}'^T).$$

Moreover, since:

$$\text{tr}(\boldsymbol{\pi}^{sT}\mathbf{X}\mathbf{X}^T\boldsymbol{\pi}^s\mathbf{x}'\mathbf{1}_{n'}^T) = \text{tr}(\mathbf{1}_{n'}^T\boldsymbol{\pi}^{sT}\mathbf{X}\mathbf{X}^T\boldsymbol{\pi}^s\mathbf{x}') = \text{tr}(\mathbf{w}^T\mathbf{X}\mathbf{X}^T\boldsymbol{\pi}^s\mathbf{x}') = \text{tr}(\boldsymbol{\pi}^{sT}\mathbf{X}\mathbf{X}^T\mathbf{w}\mathbf{x}'^T)$$

and $\text{tr}(\mathbf{w}'\mathbf{x}^T\boldsymbol{\pi}^s\mathbf{X}'\mathbf{X}'^T) = \text{tr}(\boldsymbol{\pi}^{sT}\mathbf{x}\mathbf{w}'^T\mathbf{X}'\mathbf{X}'^T)$, we can simplify the last expression to obtain:

$$\langle \mathbf{C}\boldsymbol{\pi}^s\mathbf{C}', \boldsymbol{\pi}^s \rangle = \text{tr}\big[\boldsymbol{\pi}^{sT}\mathbf{x}\mathbf{w}'^T(\mathbf{x}'\mathbf{1}_{n'}^T + \mathbf{1}_{n'}\mathbf{x}'^T) + \boldsymbol{\pi}^{sT}\mathbf{x}\mathbf{x}'^T + \mathbf{w}'\mathbf{x}^T\mathbf{w}\mathbf{x}'^T$$

$$- 4\boldsymbol{\pi}^{sT}\mathbf{x}\mathbf{w}'^T\mathbf{X}'\mathbf{X}'^T - 4\boldsymbol{\pi}^{sT}\mathbf{X}\mathbf{X}^T\mathbf{w}\mathbf{x}'^T + 4\boldsymbol{\pi}^{sT}\mathbf{X}\mathbf{X}^T\boldsymbol{\pi}^s\mathbf{X}'\mathbf{X}'^T\big].$$

Finally, we have that

$$\langle \mathbf{C}\boldsymbol{\pi}^s\mathbf{C}', \boldsymbol{\pi}^s \rangle = \text{tr}\big[\boldsymbol{\pi}^{sT}\mathbf{x}\mathbf{w}'^T\mathbf{x}'\mathbf{1}_{n'}^T + \boldsymbol{\pi}^{sT}\mathbf{x}\mathbf{w}'^T\mathbf{1}_{n'}\mathbf{x}'^T + \boldsymbol{\pi}^{sT}\mathbf{x}\mathbf{x}'^T$$

$$+ \mathbf{w}'\mathbf{x}^T\mathbf{w}\mathbf{x}'^T - 4\boldsymbol{\pi}^{sT}\mathbf{x}\mathbf{w}'^T\mathbf{X}'\mathbf{X}'^T - 4\boldsymbol{\pi}^{sT}\mathbf{X}\mathbf{X}^T\mathbf{w}\mathbf{x}'^T + 4\boldsymbol{\pi}^{sT}\mathbf{X}\mathbf{X}^T\boldsymbol{\pi}^s\mathbf{X}'\mathbf{X}'^T\big]$$

$$= \text{tr}\big[2\mathbf{w}'\mathbf{x}^T\mathbf{w}\mathbf{x}'^T + 2\boldsymbol{\pi}^{sT}\mathbf{x}\mathbf{x}'^T - 4\boldsymbol{\pi}^{sT}\mathbf{x}\mathbf{w}'^T\mathbf{X}'\mathbf{X}'^T - 4\boldsymbol{\pi}^{sT}\mathbf{X}\mathbf{X}^T\mathbf{w}\mathbf{x}'^T + 4\boldsymbol{\pi}^{sT}\mathbf{X}\mathbf{X}^T\boldsymbol{\pi}^s\mathbf{X}'\mathbf{X}'^T\big]$$

$$= 2\mathbf{w}^T\mathbf{x}\mathbf{w}'^T\mathbf{x}' + 2\langle \mathbf{x}\mathbf{x}'^T - 2\mathbf{x}\mathbf{w}^T\mathbf{X}'\mathbf{X}'^T - 2\mathbf{X}\mathbf{X}^T\mathbf{w}\mathbf{x}'^T, \boldsymbol{\pi}^s \rangle + 4\text{tr}(\boldsymbol{\pi}^{sT}\mathbf{X}\mathbf{X}^T\boldsymbol{\pi}^s\mathbf{X}'\mathbf{X}'^T).$$

The term $2\mathbf{w}^T\mathbf{x}\mathbf{w}'^T\mathbf{x}'$ is constant since it does not depend on the coupling. Also, we can verify that $c_{\mathbf{C}, \mathbf{C}'}$ does not depend on $\boldsymbol{\pi}^s$ as follows:

$$\langle c_{\mathbf{C}, \mathbf{C}'}, \boldsymbol{\pi}^s \rangle = \sum_i \|\mathbf{x}_i - \mathbf{x}_j\|_2^4 \mathbf{w}_i \mathbf{w}_j + \sum_i \|\mathbf{x}'_i - \mathbf{x}'_j\|_2^4 \mathbf{w}'_i \mathbf{w}'_j$$

implying that:

$$\langle c_{\mathbf{C}, \mathbf{C}'} - 2\mathbf{C}\boldsymbol{\pi}^s\mathbf{C}', \boldsymbol{\pi}^s \rangle = Cte - 4\langle \mathbf{x}\mathbf{x}'^T - 2\mathbf{x}\mathbf{w}^T\mathbf{X}'\mathbf{X}'^T - 2\mathbf{X}^T\mathbf{X}\mathbf{w}\mathbf{x}'^T, \boldsymbol{\pi}^s \rangle - 8\text{tr}(\boldsymbol{\pi}^{sT}\mathbf{X}\mathbf{X}^T\boldsymbol{\pi}^s\mathbf{X}'\mathbf{X}'^T).$$

We can rewrite this equation as stated in the proposition using the vec operator.

Using a standard QP form $\mathbf{c}^T\mathbf{x} + \frac{1}{2}\mathbf{x}\mathbf{Q}'\mathbf{x}^T$ with $\mathbf{c} = -4vec(\mathbf{M})$ and $\mathbf{Q}' = -4\mathbf{X}\mathbf{X}^T \otimes_K \mathbf{X}'\mathbf{X}'^T$ we see that the Hessian is negative semi-definite as the opposite of a Kronecker product of positive semi-definite matrices $\mathbf{X}\mathbf{X}^T$ and $\mathbf{X}'\mathbf{X}'^T$. $\square$

Using previous propositions we are able to prove the Proposition 3 of the paper.

**Proposition 3.** *Let* $\mathbf{C} \in \mathbb{R}^{n \times n}, \mathbf{C}' \in \mathbb{R}^{n' \times n'}$ *be any symmetric matrices, then:*

$$COOT(\mathbf{C}, \mathbf{C}', \mathbf{w}, \mathbf{w}', \mathbf{w}, \mathbf{w}') \leq GW(\mathbf{C}, \mathbf{C}', \mathbf{w}, \mathbf{w}').$$

*The converse is also true under the hypothesis of Proposition 2. In this case, if* $(\boldsymbol{\pi}_*^s, \boldsymbol{\pi}_*^v)$ *is an optimal solution of* (1), *then both* $\boldsymbol{\pi}_*^s, \boldsymbol{\pi}_*^v$ *are solutions of* (5). *Conversely, if* $\boldsymbol{\pi}_*^s$ *is an optimal solution of* (5), *then* $(\boldsymbol{\pi}_*^s, \boldsymbol{\pi}_*^s)$ *is an optimal solution for* (1) .

*Proof.* The first inequality follows from the fact that any optimal solution of the GW problem is an admissible solution for the COOT problem, hence the inequality is true by suboptimality of this optimal solution.

For the equality part, by following the same calculus as in the proof of Proposition 1, we can verify that:

$$COOT(\mathbf{C}, \mathbf{C}', \mathbf{w}, \mathbf{w}', \mathbf{w}, \mathbf{w}') = \min_{\boldsymbol{\pi}^s \in \Pi(\mathbf{w}, \mathbf{w}')} -2\text{vec}(\mathbf{M})^T \text{vec}(\boldsymbol{\pi}^s)$$
$$- 2\text{vec}(\mathbf{M})^T \text{vec}(\boldsymbol{\pi}^v) - 8\text{vec}(\boldsymbol{\pi}^s)^T \mathbf{Q}\text{vec}(\boldsymbol{\pi}^v) + Cte,$$

with $\mathbf{M}, \mathbf{Q}$ as defined in Proposition 1.

Since $\mathbf{Q}$ is negative semi-definite, we can apply Theorem 1 to prove that both problems are equivalent and lead to the same cost and that every optimal solution of GW is an optimal solution of COOT and vice versa. $\qquad\square$

## 2.3 Equivalence of DC algorithm and Frank-Wolfe algorithm for GW

Let us first recall the general algorithm used for solving COOT for arbitrary datasets.

---
**Algorithm 1** BCD for COOT
---
1: **Input:** maxIt, thd
2: $\pi_{(0)}^s \leftarrow \mathbf{w}\mathbf{w}'^T, \pi_{(0)}^v \leftarrow \mathbf{v}\mathbf{v}'^T, k \leftarrow 0$
3: **while** $k < $ maxIt **and** $err > $ thd **do**
4: $\quad \boldsymbol{\pi}_{(k)}^v \leftarrow OT(\mathbf{v}, \mathbf{v}', L(\mathbf{X}, \mathbf{X}') \otimes \boldsymbol{\pi}_{(k-1)}^s)$
5: $\quad \boldsymbol{\pi}_{(k)}^s \leftarrow OT(\mathbf{w}, \mathbf{w}', L(\mathbf{X}, \mathbf{X}') \otimes \boldsymbol{\pi}_{(k-1)}^v)$
6: $\quad err \leftarrow ||\boldsymbol{\pi}_{(k-1)}^v - \boldsymbol{\pi}_{(k)}^v||_F$
7: $\quad k \leftarrow k + 1$
---

Using Proposition 3, we know that when $\mathbf{X} = \mathbf{C}$, $\mathbf{X}' = \mathbf{C}'$ are squared Euclidean matrices, then there is an optimal solution of the form $(\boldsymbol{\pi}^*, \boldsymbol{\pi}^*)$. In this case, we can set $\boldsymbol{\pi}_{(k)}^s = \boldsymbol{\pi}_{(k)}^v$ during the iterations of Algorithm 1 to obtain an optimal solution for both COOT and GW. This reduces to Algorithm 2 that corresponds to a DC algorithm where the quadratic form is replaced by its linear upper bound.

Below, we prove that this DC algorithm for solving GW problems is equivalent to the Frank-Wolfe (FW) based algorithm presented in [6] and recalled in Algorithm 3 when $L = |\cdot|^2$ and for squared Euclidean distance matrices $\mathbf{C}', \mathbf{C}''$.

---
**Algorithm 2** DC Algorithm for COOT and GW with squared Euclidean distance matrices
---
1: **Input:** maxIt, thd
2: $\pi_{(0)}^s \leftarrow \mathbf{w}\mathbf{w}'^T$
3: **while** $k < $ maxIt **and** $err > $ thd **do**
4: $\quad \boldsymbol{\pi}_{(k)}^s \leftarrow OT(\mathbf{w}, \mathbf{w}', L(\mathbf{C}, \mathbf{C}') \otimes \boldsymbol{\pi}_{(k-1)}^s)$
5: $\quad err \leftarrow ||\boldsymbol{\pi}_{(k-1)}^s - \boldsymbol{\pi}_{(k)}^s||_F$
6: $\quad k \leftarrow k + 1$
---

---

**Algorithm 3** FW Algorithm for GW [6]

---
1: **Input:** maxIt, thd
2: $\pi^{(0)} \leftarrow \mathbf{w}\mathbf{w}'^{\top}$
3: **while** $k <$ maxIt **and** $err >$ thd **do**
4:      $\mathbf{G} \leftarrow$ Gradient from Eq. (5) *w.r.t.* $\boldsymbol{\pi}^s_{(k-1)}$
5:      $\tilde{\boldsymbol{\pi}}^s_{(k)} \leftarrow OT(\mathbf{w}, \mathbf{w}', \mathbf{G})$
6:      $\mathbf{z}_k(\tau) \leftarrow \boldsymbol{\pi}^s_{(k-1)} + \tau(\tilde{\boldsymbol{\pi}}^s_{(k)} - \boldsymbol{\pi}^s_{(k-1)})$ for $\tau \in (0, 1)$
7:      $\tau^{(k)} \leftarrow \underset{\tau \in (0,1)}{\mathrm{argmin}} \langle L(\mathbf{C}, \mathbf{C}') \otimes \mathbf{z}_k(\tau), \mathbf{z}_k(\tau) \rangle$
8:      $\boldsymbol{\pi}^s_{(k)} \leftarrow (1 - \tau^{(k)})\boldsymbol{\pi}^s_{(k-1)} + \tau^{(k)}\tilde{\boldsymbol{\pi}}^s_{(k)}$
9:      $err \leftarrow ||\boldsymbol{\pi}^s_{(k-1)} - \boldsymbol{\pi}^s_{(k)}||_F$
10:     $k \leftarrow k + 1$

---

The case when $L = |\cdot|^2$ and $\mathbf{C}, \mathbf{C}'$ are squared Euclidean distance matrices has interesting implications in practice, since in this case the resulting GW problem is a concave QP (as explained in the paper and shown in Lemma 1 of this supplementary). In [7], the authors investigated the solution to QP with *conditionally concave energies* using a FW algorithm and showed that in this case the line-search step of the FW is always 1. Moreover, as shown in Proposition 1, the GW problem can be written as a concave QP with concave energy and is minimizing *a fortiori* a conditionally concave energy. Consequently, the line-search step of the FW algorithm proposed in [6] and described in Algorithm3 always leads to an optimal line-search step of 1. In this case, the Algorithm.3 is equivalent to Algorithm 4 goven below, since $\tau^{(k)} = 1$ for all $k$.

---

**Algorithm 4** FW Algorithm for GW with squared Euclidean distance matrices

---
1: **Input:** maxIt, thd
2: $\pi^{(0)} \leftarrow \mathbf{w}\mathbf{w}'^{\top}$
3: **while** $k <$ maxIt **and** $err >$ thd **do**
4:      $\mathbf{G} \leftarrow$ Gradient from Eq. (5) *w.r.t.* $\boldsymbol{\pi}^s_{(k-1)}$
5:      $\boldsymbol{\pi}^s_{(k)} \leftarrow OT(\mathbf{w}, \mathbf{w}', \mathbf{G})$
6:      $err \leftarrow ||\boldsymbol{\pi}^s_{(k-1)} - \boldsymbol{\pi}^s_{(k)}||_F$
7:      $k \leftarrow k + 1$

---

Finally, by noticing that in the step 3 of Algorithm 4 the gradient of (5) *w.r.t* $\boldsymbol{\pi}^s_{(k-1)}$ is $2L(\mathbf{C}, \mathbf{C}') \otimes \boldsymbol{\pi}^s_{(k-1)}$, which gives the same OT solution as for the OT problem in step 3 of Algorithm 2, we can conclude that the iterations of both algorithms are equivalent.

## 2.4 Relation with Invariant OT

The objective of this part is to prove the connections between GW, COOT and InvOT [8] defined as follows:

$$\mathrm{InvOT}^L_p(\mathbf{X}, \mathbf{X}') := \min_{\boldsymbol{\pi} \in \Pi(\mathbf{w}, \mathbf{w}')} \min_{f \in \mathcal{F}_p} \langle \mathbf{M}_f, \boldsymbol{\pi} \rangle_F,$$

where $(\mathbf{M}_f)_{ij} = L(\mathbf{x}_i, f(\mathbf{x}'_j))$ and $\mathcal{F}_p$ is a space of matrices with bounded Shatten p-norms, i.e., $\mathcal{F}_p = \{\mathbf{P} \in \mathbb{R}^{d \times d} : ||\mathbf{P}||_p \leq k_p\}$.

We prove the following result.

**Proposition 4.** *Using previous notations, $L = |\cdot|^2$, $p = 2$, (i.e $\mathcal{F}_2 = \{\mathbf{P} \in \mathbb{R}^{d \times d} : ||\mathbf{P}||_F = \sqrt{d}\}$) and cosine similarities $\mathbf{C} = \mathbf{X}\mathbf{X}^T, \mathbf{C}' = \mathbf{X}'\mathbf{X}'^T$. Suppose that $\mathbf{X}'$ is $\mathbf{w}'$-whitened, i.e $\mathbf{X}'^T \mathrm{diag}(\mathbf{w}')\mathbf{X} = I$. Then, $\mathrm{InvOT}^L_2(\mathbf{X}, \mathbf{X}')$, $COOT(\mathbf{C}, \mathbf{C}')$ and $GW(\mathbf{C}, \mathbf{C}')$ are equivalent, namely any optimal coupling of one of this problem is a solution to others problems.*

In order to prove this proposition, we will need the following proposition:

**Proposition 5.** *If* $L = |\cdot|^2$, *we have that*

$$GW(\mathbf{C}, \mathbf{C}', \mathbf{w}, \mathbf{w}') = \min_{\boldsymbol{\pi}^s \in \Pi(\mathbf{w}, \mathbf{w}')} \mathbf{c}^T vec(\boldsymbol{\pi}^s) + \frac{1}{2} vec(\boldsymbol{\pi}^s)^T \mathbf{Q} vec(\boldsymbol{\pi}^s)$$

$$COOT(\mathbf{C}, \mathbf{C}', \mathbf{w}, \mathbf{w}') = \min_{\boldsymbol{\pi}^s, \boldsymbol{\pi}^v \in \Pi(\mathbf{w}, \mathbf{w}')} \frac{1}{2} \mathbf{c}^T vec(\boldsymbol{\pi}^s) + \frac{1}{2} \mathbf{c}^T vec(\boldsymbol{\pi}^v) + \frac{1}{2} vec(\boldsymbol{\pi}^s)^T \mathbf{Q} vec(\boldsymbol{\pi}^v).$$

*with* $\mathbf{Q} = -4\mathbf{C} \otimes \mathbf{C}', \mathbf{c} = vec(\mathbf{Cw1}_{n'}^T + \mathbf{1}_n \mathbf{w}' \mathbf{C}')$.

*Proof.* For GW, we refer the reader to [6, Equation 6]. For COOT we have:

$$COOT(\mathbf{C}, \mathbf{C}', \mathbf{w}, \mathbf{w}') = \min_{\boldsymbol{\pi}^s \in \Pi(\mathbf{w}, \mathbf{w}'), \boldsymbol{\pi}^v \in \Pi(\mathbf{w}, \mathbf{w}')} \langle \mathbf{L}(\mathbf{C}, \mathbf{C}') \otimes \boldsymbol{\pi}^s, \boldsymbol{\pi}^v \rangle$$

$$= \min_{\boldsymbol{\pi}^s \in \Pi(\mathbf{w}, \mathbf{w}'), \boldsymbol{\pi}^v \in \Pi(\mathbf{w}, \mathbf{w}')} \frac{1}{2} \langle \mathbf{L}(\mathbf{C}, \mathbf{C}') \otimes \boldsymbol{\pi}^s, \boldsymbol{\pi}^v \rangle + \frac{1}{2} \langle \mathbf{L}(\mathbf{C}, \mathbf{C}') \otimes \boldsymbol{\pi}^s, \boldsymbol{\pi}^v \rangle$$

$$= \min_{\boldsymbol{\pi}^s \in \Pi(\mathbf{w}, \mathbf{w}'), \boldsymbol{\pi}^v \in \Pi(\mathbf{w}, \mathbf{w}')} \frac{1}{2} \langle \mathbf{L}(\mathbf{C}, \mathbf{C}') \otimes \boldsymbol{\pi}^s, \boldsymbol{\pi}^v \rangle + \frac{1}{2} \langle \mathbf{L}(\mathbf{C}, \mathbf{C}') \otimes \boldsymbol{\pi}^v, \boldsymbol{\pi}^s \rangle$$

$$= \min_{\boldsymbol{\pi}^s \in \Pi(\mathbf{w}, \mathbf{w}'), \boldsymbol{\pi}^v \in \Pi(\mathbf{w}, \mathbf{w}')} \frac{1}{2} \langle \mathbf{Cw1}_{n'}^T + \mathbf{1}_n \mathbf{w}' \mathbf{C}', \boldsymbol{\pi}^s \rangle + \frac{1}{2} \langle \mathbf{Cw1}_{n'}^T + \mathbf{1}_n \mathbf{w}' \mathbf{C}', \boldsymbol{\pi}^v \rangle - 2 \langle \mathbf{C} \boldsymbol{\pi}^s \mathbf{C}', \boldsymbol{\pi}^v \rangle.$$

Last equality gives the desired result. $\square$

*Proof of Proposition 4.* Without loss of generality, we suppose that the columns of $\mathbf{C} = \mathbf{XX}^T, \mathbf{C}' = \mathbf{X}'\mathbf{X}'^T$ are normalized. Then, we know from [8, Lemma 4.3], that $GW(\mathbf{C}, \mathbf{C}')$ and $\text{InvOT}_2^{||\cdot||_2^2}(\mathbf{X}, \mathbf{X}')$ are equivalent. It suffices to show that $GW(\mathbf{C}, \mathbf{C}')$ and $COOT(\mathbf{C}, \mathbf{C}')$ are equivalent. By virtue of Proposition 5 the $\mathbf{Q}$ associated with the QP and BP problems of GW and COOT is $\mathbf{Q} = -4\mathbf{XX}^T \otimes_K \mathbf{X}'\mathbf{X}'^T$ which is a negative semi-definite matrix. This allows us to apply Theorem 1 to prove that $GW(\mathbf{C}, \mathbf{C}')$ and $COOT(\mathbf{C}, \mathbf{C}')$ are equivalent. $\square$

## 2.5 Relation with election isomorphism problem

This section shows that COOT approach can be used to solve the election isomorphism problem defined in [9] as follows: let $E = (C, V)$ and $E' = (C', V')$ be two elections, where $C = \{c_1, \ldots, c_m\}$ (resp. $C'$) denotes a set of candidates and $V = (v_1, \ldots, v_n)$ (resp. $V'$) denotes a set of voters, where each voter $v_i$ has a preference order, also denoted by $v_i$. The two elections $E = (C, V)$ and $E' = (C', V')$, where $|C| = |C'|$, $V = (v_1, \ldots, v_n)$, and $V' = (v'_1, \ldots, v'_n)$, are said to be isomorphic if there exists a bijection $\sigma : C \to C'$ and a permutation $\nu \in S_n$ such that $\sigma(v_i) = v'_{\nu(i)}$ for all $i \in [n]$. The authors further propose a distance underlying this problem defined as follows:

$$d\text{-ID}(E, E') = \min_{\nu \in S_n} \min_{\sigma \in \Pi(C, C')} \sum_{i=1}^{n} d\left(\sigma(v_i), v'_{\nu(i)}\right),$$

where $S_n$ denotes the set of all permutations over $\{1, \ldots, n\}$, $\Pi(C, C')$ is a set of bijections and $d$ is an arbitrary distance between preference orders. The authors of [9] compute $d\text{-ID}(E, E')$ in practice by expressing it as the following Integer Linear Programming problem over the tensor $\mathbf{P}_{ijkl} = M_{ij} N_{kl}$ where $\mathbf{M} \in \mathbb{R}^{m \times m}, \mathbf{N} \in \mathbb{R}^{n \times n}$

$$\min_{\mathbf{P}, \mathbf{N}, \mathbf{M}} \sum_{i,j,k,l} P_{k,l,i,j} |\text{pos}_{v_i}(c_k) - \text{pos}_{v'_j}(c'_l)|$$

$$\text{s.t.} \quad (\mathbf{N1}_n)_k = 1, \forall k, (\mathbf{N}^\top \mathbf{1}_n)_l = 1, \forall l \tag{6}$$

$$(\mathbf{M1}_m)_i = 1, \forall i, (\mathbf{M}^\top \mathbf{1}_m)_j = 1, \forall j$$

$$P \le N_{k,l}, P_{i,j,k,l} \le M_{i,j}, \forall i, j, k, l$$

$$\sum_{i,k} P_{i,j,k,l} = 1, \forall j, l \tag{7}$$

where $\text{pos}_{v_i}(c_k)$ denotes the position of candidate $c_k$ in the preference order of voter $v_i$. Let us now define two matrices $\mathbf{X}$ and $\mathbf{X}'$ such that $\mathbf{X}_{i,k} = \text{pos}_{v_i}(c_k)$ and $\mathbf{X}'_{j,l} = \text{pos}_{v'_j}(c'_l)$ and denote

| | | | Decaf $\rightarrow$ GoogleNet | | | |
|---|---|---|---|---|---|---|
| Domains | Baseline | CCA | KCCA | EGW | SGW | COOT |
| | | | $n_t = 1$ | | | |
| C$\rightarrow$W | 30.47$\pm$6.90 | 13.37$\pm$7.23 | 29.21$\pm$13.14 | 10.21$\pm$1.31 | 66.95$\pm$7.61 | **77.74**$\pm$4.80 |
| W$\rightarrow$C | 26.53$\pm$7.75 | 16.26$\pm$5.18 | 40.68$\pm$12.02 | 10.11$\pm$0.84 | 80.16$\pm$4.78 | **87.89**$\pm$2.65 |
| W$\rightarrow$W | 30.63$\pm$7.78 | 13.42$\pm$1.38 | 36.74$\pm$8.38 | 8.68$\pm$2.36 | 78.32$\pm$5.86 | **89.11**$\pm$2.78 |
| W$\rightarrow$A | 30.21$\pm$7.51 | 12.47$\pm$2.99 | 39.11$\pm$6.85 | 9.42$\pm$2.90 | 80.00$\pm$3.24 | **89.05**$\pm$2.84 |
| A$\rightarrow$C | 41.89$\pm$6.59 | 12.79$\pm$2.95 | 28.84$\pm$6.24 | 9.89$\pm$1.17 | 72.00$\pm$8.91 | **84.21**$\pm$3.92 |
| A$\rightarrow$W | 39.84$\pm$4.27 | 19.95$\pm$23.40 | 38.16$\pm$19.30 | 12.32$\pm$1.56 | 75.84$\pm$7.37 | **89.42**$\pm$4.24 |
| A$\rightarrow$A | 42.68$\pm$8.36 | 15.21$\pm$7.36 | 38.26$\pm$16.99 | 13.63$\pm$2.93 | 75.53$\pm$6.25 | **91.84**$\pm$2.48 |
| C$\rightarrow$C | 28.58$\pm$7.40 | 18.37$\pm$17.81 | 35.11$\pm$17.96 | 11.05$\pm$1.63 | 61.21$\pm$8.43 | **78.11**$\pm$5.77 |
| C$\rightarrow$A | 31.63$\pm$4.25 | 15.11$\pm$5.10 | 33.84$\pm$9.10 | 11.84$\pm$1.67 | 66.26$\pm$7.95 | **82.11**$\pm$2.58 |
| **Mean** | 33.61$\pm$5.77 | 15.22$\pm$2.44 | 35.55$\pm$3.98 | 10.80$\pm$1.47 | 72.92$\pm$6.37 | **85.50**$\pm$4.89 |
| | | | $n_t = 5$ | | | |
| C$\rightarrow$W | 74.27$\pm$5.53 | 14.53$\pm$7.37 | 73.27$\pm$4.99 | 11.40$\pm$1.13 | 84.00$\pm$3.99 | **85.53**$\pm$2.67 |
| W$\rightarrow$C | 90.27$\pm$2.67 | 21.13$\pm$6.85 | 85.00$\pm$3.44 | 10.60$\pm$1.05 | **95.20**$\pm$2.84 | 94.53$\pm$1.83 |
| W$\rightarrow$W | 90.93$\pm$2.50 | 15.80$\pm$3.27 | 90.67$\pm$2.95 | 9.80$\pm$2.60 | **95.40**$\pm$2.47 | 94.93$\pm$2.70 |
| W$\rightarrow$A | 90.47$\pm$2.92 | 16.67$\pm$4.85 | 87.93$\pm$2.47 | 9.80$\pm$2.68 | 95.40$\pm$1.53 | **95.80**$\pm$2.15 |
| A$\rightarrow$C | 88.33$\pm$2.33 | 15.73$\pm$4.64 | 83.13$\pm$2.84 | 10.40$\pm$1.89 | 84.47$\pm$5.81 | **91.47**$\pm$1.45 |
| A$\rightarrow$W | 88.40$\pm$3.17 | 13.60$\pm$6.25 | 87.27$\pm$2.82 | 11.87$\pm$2.40 | 87.87$\pm$4.66 | **93.00**$\pm$1.96 |
| A$\rightarrow$A | 86.20$\pm$3.08 | 14.07$\pm$2.93 | 87.00$\pm$3.48 | 14.07$\pm$1.65 | 89.80$\pm$2.58 | **92.20**$\pm$1.69 |
| C$\rightarrow$C | 75.93$\pm$4.83 | 13.13$\pm$2.98 | 70.47$\pm$3.45 | 11.13$\pm$1.52 | **85.73**$\pm$3.54 | 84.60$\pm$2.32 |
| C$\rightarrow$A | 73.47$\pm$3.62 | 15.47$\pm$6.50 | 74.13$\pm$5.42 | 11.20$\pm$2.47 | 85.07$\pm$3.26 | **87.20**$\pm$1.78 |
| **Mean** | 84.25$\pm$7.01 | 15.57$\pm$2.25 | 82.10$\pm$7.03 | 11.14$\pm$1.23 | 89.21$\pm$4.64 | **91.03**$\pm$3.97 |

Table 1: **Semi-supervised Heterogeneous Domain Adaptation** results for adaptation from Decaf to GoogleNet representations with different values of $n_t$. Note that the case $n_t$ is provided in the main paper.

by $\boldsymbol{\pi}_*^s, \boldsymbol{\pi}_*^v$ a minimizer of $\text{COOT}(\mathbf{X}, \mathbf{X}', \mathbf{1}_n/n, \mathbf{1}_n/n, \mathbf{1}_m/m, \mathbf{1}_m/m)$ with $L = |\cdot|$ and by $\mathbf{N}^*, \mathbf{M}^*$ the minimizers of problem (6), respectively.

As shown in the main paper, there exists an optimal solution for $\text{COOT}(\mathbf{X}, \mathbf{X}')$ given by permutation matrices as solutions of the Monge-Kantorovich problems for uniform distributions supported on the same number of elements. Then, one may show that the solution of the two problems coincide modulo a multiplicative factor, i.e., $\boldsymbol{\pi}_*^s = \frac{1}{n}\mathbf{N}^*$ and $\boldsymbol{\pi}_*^v = \frac{1}{m}\mathbf{M}^*$ are optimal since $|C| = |C'|$ and $|V| = |V'|$. For $\boldsymbol{\pi}_*^s$ (the same reasoning holds for $\boldsymbol{\pi}_*^v$ as well), we have that

$$(\boldsymbol{\pi}_*^s)_{ij} = \begin{cases} \frac{1}{n}, & j = \nu_i^* \\ 0, & \text{otherwise.} \end{cases}$$

where $\nu_i^*$ is a permutation of voters in the two sets. The only difference between the two solutions $\boldsymbol{\pi}_*^s$ and $\mathbf{N}^*$ thus stems from marginal constraints (6). To conclude, we note that COOT is a more general approach as it is applicable for general loss functions $L$, contrary to the Spearman distance used in [9], and generalizes to the cases where $n \neq n'$ and $m \neq m'$.

## 3 Additional experimental results

### 3.1 Complementary results for the HDA experiment

Here, we present the results for the heterogeneous domain adaptation experiment not included in the main paper due to the lack of space. Table 1 follows the same experimental protocol as in the paper but shows the two cases where $n_t = 1$ and $n_t = 5$. Table 2 and Table 3 contain the results for the adaptation from GoogleNet to Decaf features, in a semi-supervised and unsupervised scenarios, respectively Overall, the results are coherent with those from the main paper: in both settings, when $n_t = 5$, one can see that the performance differences between SGW and COOT is rather significant.

| | GoogleNet → Decaf | | | | | |
|---|---|---|---|---|---|---|
| Domains | Baseline | CCA | KCCA | EGW | SGW | COOT |
| | | | $n_t = 1$ | | | |
| C→A | 31.16±6.87 | 12.16±2.78 | 33.32±2.47 | 7.00±2.11 | 77.16±8.00 | **83.26**±5.00 |
| C→C | 30.42±3.73 | 13.74±5.29 | 32.58±9.98 | 12.47±2.81 | 76.63±8.31 | **86.21**±3.26 |
| W→A | 37.68±4.04 | 15.79±3.71 | 34.58±5.71 | 14.32±1.77 | 86.68±1.90 | **89.95**±3.43 |
| A→C | 35.95±3.89 | 15.32±8.18 | 40.16±17.54 | 13.21±3.49 | 87.89±4.03 | **90.68**±7.54 |
| A→A | 36.89±4.73 | 13.84±2.47 | 34.84±10.44 | 13.16±1.56 | 89.79±3.93 | **94.68**±2.21 |
| W→W | 32.05±4.63 | 19.89±11.82 | 36.26±21.98 | 10.00±2.59 | 84.21±4.55 | **90.42**±2.66 |
| W→C | 32.68±5.56 | 21.53±21.01 | 33.79±22.72 | 11.47±3.03 | 86.26±3.41 | **89.53**±1.92 |
| A→W | 33.84±4.75 | 16.00±7.74 | 39.32±18.94 | 11.00±4.01 | 87.21±3.67 | **91.53**±5.85 |
| C→W | 32.32±7.76 | 15.58±7.72 | 34.05±15.96 | 12.89±2.52 | 81.84±3.51 | **84.84**±5.71 |
| **Mean** | 33.67±2.45 | 15.98±2.81 | 35.43±2.50 | 11.73±2.08 | 84.19±4.43 | **89.01**±3.38 |
| | | | $n_t = 3$ | | | |
| C→A | 76.35±4.15 | 17.47±3.45 | 73.94±4.53 | 7.41±2.27 | 88.24±2.23 | **89.88**±0.94 |
| C→C | 78.94±3.61 | 18.18±3.44 | 69.94±3.51 | 14.18±3.16 | 89.71±2.25 | **91.06**±1.91 |
| W→A | 85.41±3.25 | 19.29±3.10 | 80.59±3.82 | 14.24±2.72 | 94.76±1.45 | **95.29**±2.35 |
| A→C | 89.53±4.05 | 23.18±7.17 | 80.59±6.30 | 13.88±2.69 | 93.76±2.72 | **94.76**±1.83 |
| A→A | 89.76±1.92 | 17.00±3.11 | 83.71±3.30 | 14.41±2.28 | 93.29±2.09 | **95.53**±1.45 |
| W→W | 86.65±5.07 | 21.88±4.78 | 84.65±3.67 | 9.94±2.37 | **94.88**±1.79 | 94.53±1.66 |
| W→C | 88.94±5.02 | 22.59±9.23 | 80.06±5.65 | 13.65±3.15 | **96.18**±1.15 | 95.29±2.91 |
| A→W | 90.29±1.35 | 22.35±7.00 | 87.88±2.53 | 13.88±3.60 | 94.53±1.54 | **95.35**±1.59 |
| C→W | 78.59±3.44 | 22.53±13.42 | 80.12±2.95 | 11.59±3.25 | 89.29±1.86 | **89.59**±2.22 |
| **Mean** | 84.94±5.19 | 20.50±2.34 | 80.16±5.12 | 12.58±2.31 | 92.74±2.72 | **93.48**±2.38 |
| | | | $n_t = 5$ | | | |
| C→A | 84.20±2.65 | 18.60±3.75 | 84.33±2.33 | 6.40±1.27 | **92.13**±2.61 | 91.93±2.05 |
| C→C | 85.33±2.76 | 21.80±5.91 | 78.60±2.74 | 13.47±2.00 | 91.33±2.48 | **92.27**±2.67 |
| W→A | 95.13±2.29 | 31.00±9.67 | 91.93±2.82 | 14.67±1.40 | 96.13±2.04 | **96.40**±1.84 |
| A→C | 91.67±2.60 | 21.80±4.35 | 85.33±3.27 | 13.40±3.63 | **95.47**±1.51 | 94.87±1.27 |
| A→A | 93.20±1.57 | 23.33±4.66 | 89.67±1.98 | 13.27±2.10 | **95.33**±1.07 | 95.00±1.37 |
| W→W | 95.00±2.33 | 23.80±5.48 | 92.13±1.78 | 11.20±2.58 | 96.47±1.93 | **96.67**±1.37 |
| W→C | 95.67±1.50 | 28.27±9.71 | 87.67±3.79 | 14.27±3.19 | **97.67**±1.31 | 96.93±2.25 |
| A→W | 92.13±2.36 | 22.67±3.94 | 89.20±3.14 | 11.67±2.50 | 93.60±1.40 | **94.27**±2.11 |
| C→W | 84.00±3.45 | 20.40±4.31 | 82.53±3.56 | 11.07±3.70 | 90.20±2.23 | **92.40**±1.69 |
| **Mean** | 90.70±4.57 | 23.52±3.64 | 86.82±4.26 | 12.16±2.37 | 94.26±2.42 | **94.53**±1.85 |

Table 2: **Semi-supervised Heterogeneous Domain Adaptation** results for adaptation from GoogleNet to Decaf representations with different values of $n_t$.

| | GoogleNet → Decaf | | |
|---|---|---|---|
| Domains | CCA | KCCA | EGW | COOT |
| C→A | 11.30±4.04 | 14.60±8.12 | 8.20±2.69 | **25.10**±11.52 |
| C→C | 13.35±4.32 | 17.75±10.16 | 11.90±2.99 | **37.20**±14.07 |
| W→A | 14.55±10.68 | 25.05±24.73 | 14.55±2.05 | **39.75**±17.29 |
| A→C | 13.80±6.51 | 20.70±17.94 | 16.00±2.44 | **30.25**±18.71 |
| A→A | 16.90±10.45 | 28.95±30.62 | 12.70±1.79 | **41.65**±16.66 |
| W→W | 14.50±6.72 | 24.05±19.35 | 9.55±1.77 | **36.85**±9.20 |
| W→C | 13.15±4.98 | 14.80±8.79 | 11.40±2.65 | **30.95**±17.18 |
| A→W | 10.85±4.62 | 14.40±12.36 | 12.70±2.99 | **40.85**±16.21 |
| C→W | 18.25±14.02 | 25.90±25.40 | 11.30±3.87 | **34.05**±13.82 |
| **Mean** | 14.07±2.25 | 20.69±5.22 | 12.03±2.23 | **35.18**±5.24 |

Table 3: **Unsupervised Heterogeneous Domain Adaptation** results for adaptation from GoogleNet to Decaf representations.

### 3.2 Complementary information for the co-clustering experiment

Table 4 below summarizes the characteristics of the simulated data sets used in our experiment.

| Data set | $n \times d$ | $g \times m$ | Overlapping | Proportions |
|----------|--------------|--------------|-------------|-------------|
| D1 | $600 \times 300$ | $3 \times 3$ | [+] | Equal |
| D2 | $600 \times 300$ | $3 \times 3$ | [+] | Unequal |
| D3 | $300 \times 200$ | $2 \times 4$ | [++] | Equal |
| D4 | $300 \times 300$ | $5 \times 4$ | [++] | Unequal |

Table 4: Size ($n \times d$), number of co-clusters ($g \times m$), degree of overlapping ([+] for well-separated and [++] for ill-separated co-clusters) and the proportions of co-clusters for simulated data sets.

## 4 Initialization's impact

We conducted a study regarding the convergence properties of COOT in the co-clustering application when the $\pi_s, \pi_v$ and $X_c$ are initialized randomly over 100 trials. This leads to a certain variance in the obtained value of the COOT distance as expected when solving a non-convex problem. The obtained CCEs remain largely in line with the obtained results even for different random initializations.

| Data set | Characteristics | | | |
|----------|-----------------|--|--|--|
| | Runtime(s) | BCD #iter. (COOT+$\mathbf{X}_c$) | BCD #iter. (COOT) | COOT value |
| D1 | 4.72±6 | 21.5±24.57 | 3.16±0.37 | 0.46±0.25 |
| D2 | 0.64±0.81 | 9.77±11.53 | 3.4±0.58 | 1.35±0.16 |
| D3 | 0.95±1.55 | 8.47±11.11 | 3.01±0.1 | 2.52±0.24 |
| D4 | 6.27±5.13 | 33.15±23.75 | 4.21±0.41 | 0.06±0.005 |

Table 5: Mean (± standard-deviation) of different runtime characteristics of COOT.