[Reviews · NeurIPS 2020]

Review 1

Summary and Contributions: This paper introduces a method based on optimal transport, COOT, to compare distributions supported on different vector spaces. It consists in simultaneously learning two transport plans, one for samples and one for features. The authors provide an algorithm to compute COOT which can be adapted to entropic regularisation, and discuss the relation between COOT and Gromov-Wasserstein (GW). Finally, they illustrate COOT on heterogeneous domain adaptation and co-clustering (i.e., clustering of both samples and features) tasks.

Strengths: The problem of simultaneously matching samples and features across different domains is well-motivated (as stated by the authors, it notably has applications to embedding alignment in e.g. NLP, and domain adaptation) and the proposed method is convincing and novel. It adapts OT and GW frameworks in a natural manner. The fact that COOT still defines a distance (in a permutational sense) shows that COOT is theoretically well-grounded. The links established with GW by the authors are informative. The illustrations and experiments showcase COOT convincingly.

Weaknesses: The main weakness of this work lies in the computational cost of COOT, as Algorithm 1 relies on iteratively solving OT or matrix balancing problems (in the case of entropic regularization). However, this is arguably inherent to the problem it tackles, as GW also suffers from high costs, and the authors provide clear information on the complexity of their method. Adding plots (e.g. in the supplemental) showing the empirical behaviour of Algorithm 1 in different settings (values of n, d, and regularization) would help assessing how big an issue this complexity is.

Correctness: I have not checked all proofs in detail but the results and the experimental methodology seem correct.

Clarity: I have found the paper very clearly written (at least for someone having some knowledge on OT and GW). One point that could be clarified however is what loss L is used in the experiments. I assume it is quadratic - this is clearly stated in the co-clustering experiment description but I couldn’t find the information for HDA. Likewise, what regularization is exactly used? If I understand well, entropic regularisation for both features and samples in co-clustering, but only for features in HDA? If so, could the authors discuss these different choices?

Relation to Prior Work: The relation to prior work is clearly discussed.

Reproducibility: Yes

Additional Feedback: (i) In Proposition 1, the authors show that if L is an Lp norm, then COOT is positive, symmetric and satisfies the triangle inequality, and further that if n=n’ and d=d’ COOT(X, X’) is 0 iff X = X’ up to permutation. Then, l. 134-135 they say that when n≠n’ or d≠d’, the identity of indiscernibles cannot be proven, but (l.60 in the supplemental) that COOT is then > 0. But arguably, when n≠n’ or d≠d’, wouldn’t X and X’ be different and hence the identity of indiscernibles be still satisfied, which would show that COOT is still a distance (the point being that “X=X’” in a permutational sense implies that n=n’ and d=d’)? (ii) In Figure 1, the authors illustrate the sample and feature plans between MNIST and USPS datasets. However, the feature mapping is a little hard to read. It would be interesting to illustrate feature mappings between two datasets which have clearly interpretable features (e.g. physical quantities), to see whether features having similar roles are indeed matched to one another. Less importantly, although I am not a grammar expert it seems that in some cases the usage of possessives is a bit unusual (e.g. l.28 “correspondences’ estimation” could be “correspondence estimation”, l.213 “features’ mapping” -> “feature mappings”, etc.) ########### POST-REBUTTAL ############# Thanks to the authors for answering my points. I have read all reviews and responses, and agree with other reviewers that empirical convergence is the main issue. As the authors have provided some content on that subject in their response, I will maintain my score.


Review 2

Summary and Contributions: This work proposed a new formulation of Optimal transport (CO-OT) to compare distributions in different metric spaces with potentially different dimensions. Given two datasets, the main idea is to align both the samples and the features by optimizing over a pair of transportation maps. An alternating optimization algorithm is proposed to solve this problem (performing several classical OT subproblems) and several numerical experiments are shown to illustrate the intuition behind the double-assignment (MNIST-USPS) and the ability of CO-OT to out-perform state of the art methods in domain adaptation and co-clustering applications.

Strengths: - CO-OT has a simple and intuitive formulation and can be solved using previously known algorithms iteratively. - It provides a generalization for Gromov-Wasserstein that leverages the raw data instead of kernel representations of the data. - The experiments show a significant improvement in accuracy compared to other methods

Weaknesses: - Small discussion on computational aspects - Lack of context of the obtained results

Correctness: - Except some obscure settings in the experiments, the main claims of this paper are sound.

Clarity: Well written, well illustrated.

Relation to Prior Work: Several methods are discussed throughout the paper and compared with in the experiments. Authors also made important connections of CO-OT with some relevant proposals in the literature (Gromov-W, InvOT).

Reproducibility: Yes

Additional Feedback: ####### POST REBUTTAL I find the authors' response very solid. My questions were answered convincingly, and I would very much like to see the 3rd paragraph of the rebuttal in the final version of the paper with the additional experiments. ################# Overall this is a good paper; CO-OT is well motivated and illustrated using the MNIST / USPS example. The positioning is clear with respect to Gromov-Wasserstein and the algorithmic part amounts to nested loops of the simplex / Sinkhorn algorithm. 1) In L121: the statement "COOT provides a tight convex relaxation of BAP" is true in terms of the constraint set but still the objective function is not jointly convex in the pair (pi_s, pi_v) ? Did authors try different initializations of Alg 1. to see how sensitive it is, specially for co-Clustering methods when minimizing over X? 2) Given the (at-least) quadratic complexity of the inner OT-problems, computing COOT would be problematic if the alternating opt takes several iterations to converge. How many iterations were necessary to reach convergence ? It could be useful to have a plot showing the decrease of the loss function. A quick look at the code shows that Alg 1 could be accelerated by keeping track of the previous dual variables to warmstart Sinkhorn's algorithm. 3) The accuracy scores obtained by EGW in all experiments tables (even those in the appendix) are no better than chance ! Perhaps the choice of eps was not adequate here ? Several hyperparameters were fixed once and for all L249-252 with [16] as a ref, but in [16] EGW performed significantly better and was even competing with SGW. 4) minor: latex typo in L296


Review 3

Summary and Contributions: The paper proposes a new OT distance called COOT which simultaneously considers differences of samples as well as their features of two interested distributions. The experimental studies on heterogeneous domain adaptation and bi-clustering problems have shown the usefulness of the proposed distance.

Strengths: The proposed OT distance is useful in problems when ambient data spaces of two distributions are different. The authors also provide the optimization algorithm to compute the distance (and transportation plans). Applications of COOT for heterogeneous domain adaptation have shown the performance improvements over the state-of-the-art methods.

Weaknesses: Though the usefulness of COOT has shown via two applications experimentally, theoretical grounding for COOT is missing. In COOT-clustering application, authors do not show some quantitative results on real-world datasets such as Olivetti Face and MovieLens. ==== Update after rebuttal==== I appreciate the feedbacks from authors. However, I am still not satisfied with the quantitative results on real-world datasets as synthetic datasets are supposed for validation (the correctness) not for comparing with the baseline methods.

Correctness: Claims in three propositions look correct without further checking.

Clarity: The paper is easy to follow however, the motivation for the COOT needs be elaborated.

Relation to Prior Work: Authors have discussed connections to other OT distances especially the Gromov-Wasserstein (GW) distance.

Reproducibility: Yes

Additional Feedback: In line 39, authors claim that "our new formulation includes GW as a special case", can authors show at which conditions, COOT becomes GW?


Review 4

Summary and Contributions: The paper addresses the problem of solving optimal transport in a context where the two spaces are of different dimensions and the number of samples may be different. Therefore it presents an alternative to Gromov-Wasserstein. Unlike GW, which relies on the distributions of distances between pairs of points in each space, COOT makes use of the features, by finding two couplings instead of one: one between samples and one between features. Even though the problem turns out to be NP-hard, one can solve it approximately via block coordinate descent, which basically boils down to alternatively solve OT for one coupling, and then for the other until it converges. The main contribution of this paper is to propose this new approach to solve two couplings instead of one and a simple algorithm grounded on OT.

Strengths: To begin with, the paper is well-written and easy to understand despite the theoretical complexity of the problem. Then, I have appreciated the effort put into linking the paper with previous work and more in general the presence of substantial references to the literature to support their various claims. In my opinion, the proposed algorithm shines by its simplicity, which basically alternates between two OT problems. Eventually, the illustration of the method is quite convincing and the experimental validation supports their claims.

Weaknesses: Although the complexity analysis is given to the reader, I think that details about empirical speed of convergence of the algorithm and its memory limitations are lacking. Even though Sinkhorn algorithm can be qualified of "light speed" and even though its complexity is quadratic, alternating many times a sequence of two Sinkhorn algorithms might in the end be costly in time and memory. I would have liked to know how long it takes to run on MNIST / USPS, how many iterations of Sinkhorn for each inner OT and how many outer iterations are needed. I am asking myself if that is the reason why in the HDA experiment, the authors are only taking 20 samples per class, ending up with n=n'=200 instead of using a bigger set. What is the limitation here and can other methods support more samples ? In which case it might also be unfair to limit other methods to 200 samples because COOT cannot handle more than this, if that is the reason for this choice. If that is indeed the case, one possible way to deal with this would be to use a subset of samples to run COOT in a first step that is only meant to find the features transport \pi^v, and then use all the transported samples (by \pi^v) to find the transport \pi^s in the common feature space. About the face dataset, could the same experiment be run on CelebA ? Or are the images too big for this ? In other words, what can a reader expect from COOT in terms of ranges of n, n', d, d' ?

Correctness: I believe so. With this one question about the empirical setup in HDA that restricts the number of samples to 20 per class.

Clarity: The paper is well written and very clear. It is a good read.

Relation to Prior Work: I am not an expert in the field, but I have enjoyed having a rich bibliography and I felt like most of the claims were supported by citations and that the experimental validation compares itself to many previous works. Also I am not aware of any other work trying to work on the samples and the features at the same time.

Reproducibility: Yes

Additional Feedback: ########### POST-REBUTTAL ############# I would like to thank the authors for answering my questions in their rebuttal. I find the paper and the rebuttal convincing and I am maintaining my score. ####################################### A small error on line 85, the authors refers to paragraph 2.2 as explaining the algorithm, however, the algorithm is described in paragraph 2.3. Figure 1 is very interesting but it takes time to parse at first glance. I feel like some examples from figure 2 and 3 in the supplementary materials are easier to understand and would be complementary. Also even if I acknowledge the effort put into solving different tasks to illustrate the method, the last one about movie ratings does not really bring much to the paper in my opinion, and I think this space could be used to discuss practical running time / memory usage instead or could be used to bring some interesting results / proof from the supplementary material, that is very complete. Eventually, I am wondering if it would be possible to add more constraints on \pi^v without breaking the simplicity of the method. For example in the MNIST -> USPS example, we do expect some piece-wise continuity, meaning that the nearby pixel locations of MNIST should be also close in USPS. The features order is not completely random in this specific case and this structure in preserved in both spaces.

[Author Response · NeurIPS 2020]

We start by thanking the reviewers for their detailed reviews and comments that will help improving the final version of the paper. Below, we address the different remarks made by the reviewers.

### ——- Theoretical aspects ——-

(**R1: COOT is a distance**) COOT is a distance in general in the permutational sense. When $n \neq n'$, $d \neq d'$ we have indeed COOT $> 0$. That is what we awkwardly meant by "identity of indiscernibles cannot be proven". We will clarify it. (**R2: COOT as a tight convex relaxation of BAP**) The reviewer is correct: the problem is not jointly convex in $(\pi_s, \pi_v)$. By convex relaxation, we target specifically the set of constraints but keeping the latter tight as we still recover a solution of the original BAP problem. (**R3: COOT and GW**) COOT includes GW as special case and both are the same when the problem is concave (*e.g* when $\mathbf{X}, \mathbf{X}'$ are squared Euclidean distance matrices) as discussed in Prop 3. (**R3: missing theoretical grounding**) As mentioned in the conclusion, the continuous formulation of COOT is indeed of high interest. We chose to focus on studying its discrete version with use-cases that are more relevant for the ML community. We hope this work will pave the way for more theoretical studies on this particular novel instance of OT problem.

### ——- Experimental settings ——-

(**R1: choice of L**) In all experiments we found $L = |\cdot|^2$ to be efficient, but we agree that a deeper analysis on its choice can be relevant for future works. (**R1: which regularization?**) For co-clustering, we use entropic regularization on features and samples to obtain soft clustering assignments. For HDA experiment, we use entropic regularization on features only as the number of samples is relatively low, and following practices of OT in domain adaptation where the entropic regularization proved to be efficient for handling such cases. (**R2: low scores for EGW**) We confirm the low scores for EGW. While we acknowledge that the choice of the hyperparameter might not be optimal, we observed that the score on the test set remained low for most of the values tested. Contrary to [16], the features here are more high-dimensional (DeCAF and GoogleNet). We suspect that EGW cannot handle the cases where $n$ is low and $d$ is large. (**R3: scores on Olivetti and MovieLens**) Our goal for these two datasets was to highlight qualitatively the COOT's ability to find meaningful solutions to a quantization problem. A quantitative study of COOT w.r.t. other co-clustering baselines is given on simulated datasets with known ground truth. (**R4: 20 samples per class in HDA**) This is the classical setting for this experiment. It was introduced in Yan et al. "Learning Discriminative Correlation Subspace for HDA", IJCAI'17 and used in [16]. We will add this citation.

### ——- Timings & Computational complexities ——-

We will detail both time and memory complexities of COOT in the final version.

MNIST/USPS. Optimization value
- No reg. (total time=46s)
- Entropic reg. (total time=40s)

(**R1&R2&R4: runtime details**) For an 1e-20 precision, the obtained runtime characteristics for the MNIST/USPS example and co-clustering experiments are given in figure on the left and table below. As one can see, the number of iterations for the BCD do not generally exceed 20. This means that the complexity of COOT mostly depends on the complexity of the used OT solver. Also, for HDA the timing of COOT is comparable to the one of SGW ($\sim 10s$), but superior to the one of KCCA ($\sim 0.1s$) to solve for one pair. We will include a more general study on simulated data with different values of $n$ and $d$, as suggested by R1, in the Supplementary material.

(**R2: initialization's impact**) We conducted a study regarding the convergence properties of COOT in the co-clustering application when the $\pi_s, \pi_v$ and $X_c$ are initialized randomly over 100 trials. This leads to a certain variance in the obtained value of the COOT distance as expected when solving a non-convex problem.

| Data set | Characteristics | | | |
|---|---|---|---|---|
| | Runtime(s) | BCD #iter. (COOT+$X_c$) | BCD #iter. (COOT) | COOT value |
| D1 | 4.72±6 | 21.5±24.57 | 3.16±0.37 | 0.46±0.25 |
| D2 | 0.64±0.81 | 9.77±11.53 | 3.4±0.58 | 1.35±0.16 |
| D3 | 0.95±1.55 | 8.47±11.11 | 3.01±0.1 | 2.52±0.24 |
| D4 | 6.27±5.13 | 33.15±23.75 | 4.21±0.41 | 0.06±0.005 |

Table 1: Mean ($\pm$ standard-deviation) of different runtime characteristics of COOT.

The obtained CCEs remain largely in line with the obtained results even for different random initializations. The quantitative results will be included in the paper, following the recommendation of R4. (**R4: scalability of COOT**) While our current implementation relies on solvers that compute couplings solutions to the primal OT problem (near linear time complexity for entropic regularization [23] but with a quadratic memory overhead), stochastic solvers working solely with dual variables could be used to efficiently deal with large datasets such as CelebA (eg, with neural networks as dual potentials). As suggested by R2 (thanks for the insights), warm starting the solvers inbetween BCD iterations can also accelerate our code and is an exciting avenue for scaling up COOT computation that we are currently working on.

[Meta-Review · NeurIPS 2020]

Reviewers found the results of the paper significant and of interest. Authors should address the comments raised by the reviewers in the final draft.